# ON THE MARKOV PROPERTY OF NEURAL ALGORITHMIC REASONING: ANALYSES AND METHODS

**Montgomery Bohde,*** **Meng Liu,*** **Alexandra Saxton, Shuiwang Ji**
Department of Computer Science & Engineering
Texas A&M University
College Station, TX 77843, USA
`{mbohde,mengliu,allie.saxton,sji}@tamu.edu`

## ABSTRACT

Neural algorithmic reasoning is an emerging research direction that endows neural networks with the ability to mimic algorithmic executions step-by-step. A common paradigm in existing designs involves the use of historical embeddings in predicting the results of future execution steps. Our observation in this work is that such historical dependence intrinsically contradicts the Markov nature of algorithmic reasoning tasks. Based on this motivation, we present our ForgetNet, which does not use historical embeddings and thus is consistent with the Markov nature of the tasks. To address challenges in training ForgetNet at early stages, we further introduce G-ForgetNet, which uses a gating mechanism to allow for the selective integration of historical embeddings. Such an enhanced capability provides valuable computational pathways during the model's early training phase. Our extensive experiments, based on the CLRS-30 algorithmic reasoning benchmark, demonstrate that both ForgetNet and G-ForgetNet achieve better generalization capability than existing methods. Furthermore, we investigate the behavior of the gating mechanism, highlighting its degree of alignment with our intuitions and its effectiveness for robust performance. Our code is publicly available at https://github.com/divelab/ForgetNet.

## 1 INTRODUCTION

Neural algorithmic reasoning stands at the intersection of neural networks and classical algorithm research. It involves training neural networks to reason like classical algorithms, typically through learning to execute step-by-step algorithmic operations (Veličković & Blundell, 2021; Veličković et al., 2022a). Since classical algorithms inherently possess the power to generalize across inputs of varying sizes and act as "building blocks" for complicated reasoning pathways, learning to mimic algorithmic execution can confirm and amplify the generalization and reasoning abilities of neural network models (Xu et al., 2020; Deac et al., 2021; Numeroso et al., 2023; Veličković et al., 2022b).

Existing works based on the CLRS-30 benchmark (Veličković et al., 2022a) have demonstrated the effectiveness of mimicking algorithmic operations in high-dimensional latent space (Veličković et al., 2020b; Georgiev & Lió, 2020; Veličković et al., 2020a; 2022a; Ibarz et al., 2022; Diao & Loynd, 2023). As detailed in Section 3.1, they typically employ an encoder-processor-decoder framework to learn the step-by-step execution of algorithms. At each step, the current algorithm state is first embedded in a high-dimensional latent space via the encoder. The embedding is then given to the processor to perform one step of computation in the latent space. The processed embeddings are then decoded to predict the updated algorithm state, namely `hints`. Within this paradigm, a common practice is to use historical embeddings in the current execution step. Our insight in this work is that such historical dependence contradicts the intrinsic nature of classical algorithms.

Our work is motivated by the Markov property of algorithmic reasoning tasks; that is, the present state is sufficient to fully determine the execution output of the current step. This observation led us to investigate if the use of historical embeddings in the existing paradigm is indeed useful as it does

---

*Equal contribution

not align with the underlying Markov property. Such a misalignment introduces noise, thus hindering the model's generalization ability, especially in out-of-distribution scenarios. To be consistent with the Markov property, we present ForgetNet, which removes the dependency on historical embeddings and explicitly embraces the Markov nature of algorithmic reasoning tasks. Such a modification, while simple, fundamentally realigns the computational graph of the neural model with the inherent structure of algorithmic processes. We observe that, although ForgetNet shows improvements across a wide range of tasks, training such models may be challenging due to inaccurate intermediate state predictions, especially at the early stages of training. To improve training, we further enhance our design with G-ForgetNet, in which a regularized gating mechanism is introduced in order to align with the Markov property during testing while still allowing for beneficial computational pathways during training.

Our extensive experimental evaluations on the widely used CLRS-30 algorithmic reasoning benchmark demonstrate that both ForgetNet and G-ForgetNet outperform established baselines. In particular, G-ForgetNet achieves robust and promising performance in many different tasks, showing the benefit of the proposed gating mechanism. Further in-depth analyses of the training dynamics and gate behavior shed light on our understanding of the advantages of the proposed approaches. Overall, the findings in this work demonstrate the importance of aligning model design with the underlying Markov nature to achieve better generalization performance in neural algorithmic reasoning tasks.

## 2 RELATED WORK

Equipping neural networks with algorithmic reasoning abilities has gained increasing attention in recent research. Early attempts (Zaremba & Sutskever, 2014; Graves et al., 2014; Kaiser & Sutskever, 2015; Graves et al., 2016; Joulin & Mikolov, 2015) in this direction typically use recurrent neural networks with augmented memory mechanisms to mimic algorithms, showing that neural models could learn algorithmic patterns from data. With the use of graph-based representations, graph neural networks (GNNs) (Gilmer et al., 2017; Battaglia et al., 2018) can be applied naturally to algorithmic reasoning tasks (Veličković et al., 2020b; Georgiev & Lió, 2020; Veličković et al., 2020a; Xu et al., 2020; Yan et al., 2020; Dudzik & Veličković, 2022; Dwivedi et al., 2023). Intuitively, the message passing schema in various GNNs can naturally model the propagation and iterative nature of many classical algorithms. Recently, Veličković & Blundell (2021) outlines a blueprint for neural algorithmic reasoning, proposing a general encoder-processor-decoder framework. The framework, trained in algorithmic tasks, produces processors with potential applicability in real-world applications. Such generalization and transferable reasoning capabilities have been showcased in a few prior studies (Deac et al., 2021; Numeroso et al., 2023; Veličković et al., 2022b; Beurer-Kellner et al., 2022). In addition, Xhonneux et al. (2021); Ibarz et al. (2022) have explored the generalization ability of the processor across multiple algorithms.

To provide a comprehensive testbed for algorithm reasoning tasks, Veličković et al. (2022a) presents the CLRS-30 benchmark, which covers 30 classical algorithms that span sorting, searching, dynamic programming, geometry, graphs, and strings (Cormen et al., 2022). The CLRS-30 benchmark is known for its out-of-distribution (OOD) testing setup (Mahdavi et al., 2022), where the input size of testing samples is much larger than those during training. Such a setup provides a rigorous test for the generalization capability of models, serving as a standard testbed for algorithmic reasoning tasks. In the benchmark study, multiple representative neural models, including Deep Sets (Zaheer et al., 2017), GAT (Veličković et al., 2018), MPNN (Gilmer et al., 2017), PGN (Veličković et al., 2020a), and Memnet (Sukhbaatar et al., 2015), have been evaluated as the processor network within the encoder-processor-decoder framework. Based on this benchmark, Ibarz et al. (2022) further proposes Triplet-GMPNN, which employs a triplet message passing schema (Dudzik & Veličković, 2022) and multiple improvements for the training stability of the encoder-processor-decoder framework. Recently, Bevilacqua et al. (2023) proposes an additional self-supervised objective to learn similar representations for inputs that result in identical intermediate computation. a common design in the above methods is the incorporation of historical embeddings into current execution steps. In this work, we highlight that such historical dependence poses a misalignment with the Markov nature of the algorithmic execution. This insight motivates our proposed ForgetNet and its enhanced version G-ForgetNet, which more faithfully align with the Markov nature by reconsidering the use of historical embeddings.

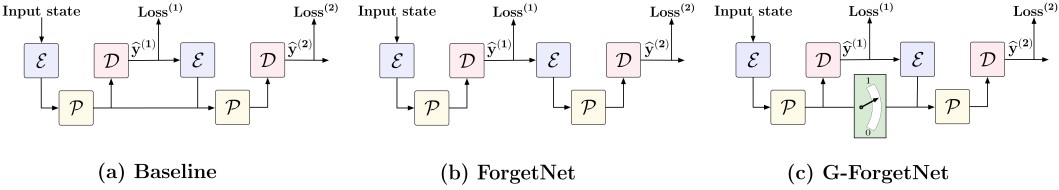

Figure 1: An illustration of (a) the baseline, (b) ForgetNet, and (c) G-ForgetNet methods. $\mathcal{E}$, $\mathcal{P}$, and $\mathcal{D}$ represent the encoder, processor, and decoder module, respectively.

## 3 ANALYSES ON THE MARKOV PROPERTY

In this section, we first recap the existing encoder-processor-decoder paradigm on the algorithmic reasoning tasks given in the CLRS-30 benchmark. Then, we emphasize the Markov characteristic of algorithmic executions. In addition, we highlight the existing misalignment between the use of historical embeddings and this Markov property. Motivated by this observation, we present ForgetNet, which removes such historical embeddings to achieve a closer alignment with the task's nature. An empirical study validates that ForgetNet achieves better generalization capability.

### 3.1 ENCODER-PROCESSOR-DECODER PARADIGM

Following prior research, we consider the algorithmic reasoning tasks as formulated in the CLRS-30 benchmark (Veličković et al., 2022a). For a certain algorithm, a single execution trajectory serves as a data sample, which is composed of the `input`, `output`, and `hints`. Here, `hints` are a time series of intermediate states of the algorithm execution. Typically, a data sample is represented as a graph with $n$ nodes, where $n$ reflects the size of a particular sample. For example, in sorting algorithms, elements in the input list of length $n$ are denoted as $n$ nodes. With such a graph representation, the `input`, `output`, and `hints` at a particular time step are either located in node-level, edge-level, or graph-level features. As detailed in Veličković et al. (2022a), there are five possible types of features, including `scalar`, `categorical`, `mask`, `mask_one`, and `pointer`, each accompanied by its encoding/decoding strategies and associated loss functions.

Let us denote the node-level, edge-level, and graph-level features at time step $t$ as $\{\boldsymbol{x}_i^{(t)}\}$, $\{\boldsymbol{e}_{ij}^{(t)}\}$, and $\boldsymbol{g}^{(t)}$, respectively. Here, $i$ indicates the node index and $ij$ specifies the index of the edge between node $i$ and $j$. Note that in addition to the `input`, `hints` are also included in these features when they are available. Most existing neural algorithmic learners (Veličković et al., 2020b; Georgiev & Lió, 2020; Veličković et al., 2020a; 2022a; Ibarz et al., 2022; Diao & Loynd, 2023) adopt the encoder-processor-decoder paradigm (Hamrick et al., 2018). Specifically, at each time step $t$, the encoder first embeds the current features into high-dimensional representations as

$$\bar{\boldsymbol{x}}_i^{(t)} = f_n\left(\boldsymbol{x}_i^{(t)}\right), \quad \bar{\boldsymbol{e}}_{ij}^{(t)} = f_e\left(\boldsymbol{e}_{ij}^{(t)}\right), \quad \bar{\boldsymbol{g}}^{(t)} = f_g\left(\boldsymbol{g}^{(t)}\right). \tag{1}$$

Here, $f_n(\cdot)$, $f_e(\cdot)$, and $f_g(\cdot)$ are the encoder layers, typically parameterized as linear layers. The embeddings are then fed into a processor, which is parameterized as a graph neural network $f_{\text{GNN}}(\cdot)$, to perform one step of computation. The processor can be formulated as

$$\boldsymbol{z}_i^{(t)} = \left[\bar{\boldsymbol{x}}_i^{(t)}, \boldsymbol{h}_i^{(t-1)}\right], \quad \{\boldsymbol{h}_i^{(t)}\} = f_{\text{GNN}}\left(\{\boldsymbol{z}_i^{(t)}\}, \{\bar{\boldsymbol{e}}_{ij}^{(t)}\}, \bar{\boldsymbol{g}}^{(t)}\right), \tag{2}$$

where $[\cdot]$ denotes concatenation. It is worth noting that the processed node embeddings from the previous step, $\{\boldsymbol{h}_i^{(t-1)}\}$, are used at each time step $t$. Initially, $\boldsymbol{h}_i^{(0)} = \boldsymbol{0}$ for all nodes. Subsequently, the decoder, a linear model, uses the processed node embeddings $\{\boldsymbol{h}_i^{(t)}\}$ to either predict the `hints` for the next time step, or the `output` if it is at the final time step. Note that the encoder and decoder should be task-tailored based on the feature types in the particular task. Additionally, the learnable parameters of all neural modules are shared over time steps. To train the described encoder-processor-decoder model, the loss is calculated based on the decoded `hints` at every step and the `output` at the end.

Figure 2: Illustration of the execution steps in insertion sort. The top row represents the intermediate states, while the bottom row shows the corresponding partially sorted lists. At a specific step, the present state, denoted as the `hints`, includes the current order (the black pointers), the recently inserted element (the green pointer), and the current iterator (the blue pointer). The present state can fully determine the next intermediate state. The figure is adapted from Veličković et al. (2022a).

During training, either the ground truth `hints` or the `hints` predicted from the previous step can be fed into the encoder, depending on whether teacher forcing is used. During inference, the step-by-step `hints` are not available, and the encoder always receives the predicted `hints` from the previous step. In the benchmark study by Veličković et al. (2022a), the ground truth `hints` are used with $50\%$ probability during training, given that the training process would become unstable without teacher forcing. While using the actual `hints` can stabilize training, it introduces discrepancies between training and inference modes. Recently, Ibarz et al. (2022) proposes several techniques to improve training stability, such as using soft hint prediction, specific initialization, and gradient clipping tricks. More importantly, it demonstrates that, with such training stability, it is possible to completely remove teacher forcing and enforce the model to rely on the `hints` predicted from the previous step, thus aligning the training with inference and achieving better performance. Therefore, as illustrated in Figure 1 (a), our study in this work specifically adopts and builds on this pipeline that operates without relying on teacher forcing.

## 3.2 ALGORITHMIC NECESSITY OF HISTORICAL EMBEDDINGS

**Markov nature of algorithmic executions.** The Markov property refers to the principle that future states depend only on the current state and not on the sequence of states that preceded it. It is important to note that such fundamental property holds in the context of algorithmic reasoning tasks formulated in the CLRS-30 benchmark because the entire algorithm state is given in each `hints`. To be specific, within an algorithm's sequential execution, the state at a time step $t$ encompasses all necessary information to unambiguously determine the state at the subsequent time step $t + 1$, preventing the need to refer to any states preceding time step $t$. Let us take the insertion sort in Figure 2 as an example. At any specific step, the intermediate state, represented as the `hints`, completely determines the algorithmic execution output of that particular step, *i.e.*, the next intermediate state.

**Misalignment with the use of historical embeddings.** Given the Markov nature of the task, we revisit the necessity of using historical embeddings in the existing paradigm for algorithm reasoning. As described in Section 3.1, a prevalent practice in the existing encoder-processor-decoder framework is the incorporation of historical embeddings from previous steps into the current processor input. This practice, which might seem to naturally borrow from design principles in graph neural networks (GNNs) and recurrent neural networks (RNNs), intends to capture and propagate potentially relevant information across time steps. However, it intrinsically contradicts the Markov nature of the task as highlighted above. Given the Markov property of tasks within the CLRS-30 benchmark, the progression of the algorithm should depend solely on the current state, given by the current `hints`. The incorporation of historical embeddings from previous steps, while seemingly advantageous, might inadvertently add unnecessary complexity to the model. Such an addition not only complicates the model architecture but also introduces potential discrepancies and noise that might misguide our neural learners away from the desired algorithmic trajectory, consequently compromising the generalization ability.

**ForgetNet: removing the use of historical embeddings.** As studied by Xu et al. (2020), it is easier for neural networks to learn reasoning tasks where the computational graph of the neural network aligns with the algorithmic structure of the task since the network only needs to learn simple algorithm steps. Motivated by this intuition and the identified misalignment between the use of historical embeddings and the Markov nature of neural algorithmic reasoning tasks, we suggest removing the use of historical embeddings to align the computational graph of the neural model

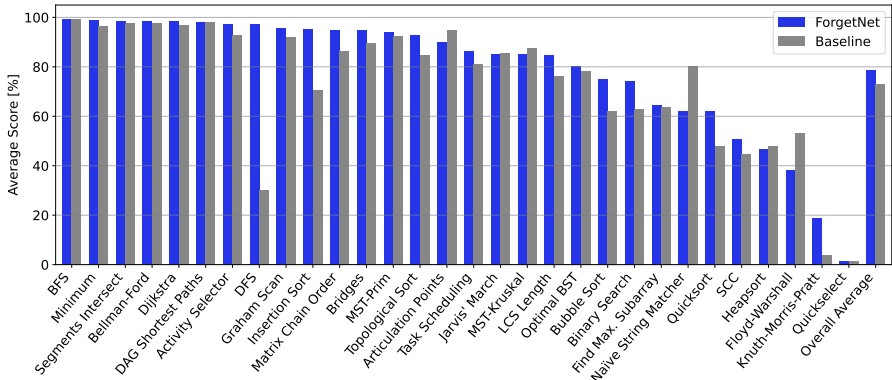

Figure 3: Comparison between ForgetNet and the baseline. Reported results are the average of 10 runs with random seeds. Numerical results can be found in Table 1.

with the task's Markov nature. Specifically, following the notation in Eq. (2), we remove the use of $\{h_i^{(t-1)}\}$ and only use the encoded node embeddings $\{\bar{x}_i^{(t)}\}$ as the input node embeddings for the processor. Formally, the processor as in Eq. (2) is replaced with

$$\{h_i^{(t)}\} = f_{\text{GNN}}\left(\{\bar{x}_i^{(t)}\}, \{\bar{e}_{ij}^{(t)}\}, \bar{g}^{(t)}\right). \tag{3}$$

While the modification of the model architecture seems simple, it non-trivially enables the updated model to have a direct and coherent alignment with the underlying Markov nature of the neural algorithmic reasoning task. The parameterized processor can thus focus on learning the one-step execution of the algorithm, without the potential discrepancies introduced by using historical embeddings. This new streamlined framework, as illustrated in Figure 1 (b), is termed ForgetNet.

**Empirical validation.** To verify our insight, using the full set of algorithms from the CLRS-30 benchmark, we train our ForgetNet alongside the existing architecture as a baseline (*i.e.*, Figure 1 (b) *vs.* Figure 1 (a)). The only difference between these two models is that the historical embeddings are removed in ForgetNet. Using the standard OOD splits in the CLRS-30 benchmark, we perform 10 runs for each model on each algorithm task with a single set of hyperparameters. As demonstrated in Figure 3, ForgetNet improves the performance over the baseline across 23/30 algorithmic reasoning tasks. The improvements brought by removing historical embeddings are quite significant on several tasks. For example, the absolute margins of improvement on DFS, insertion sort, and bubble sort are $66.79\%$, $24.57\%$, and $13.19\%$ respectively. By focusing purely on the relevant signals at the current step, ForgetNet can generalize better to OOD testing samples, fitting more useful signals for improved performance. In Appendix B.1, we further evaluate the performance of ForgetNet on the multi-task setup following Ibarz et al. (2022). These empirical studies directly verify our insight that it is effective to explicitly enforce the Markov property in neural algorithmic learners.

## 4 IMPROVED TRAINING VIA ADAPTIVE ALIGNMENT

In this section, we first identify the limitation of completely removing historical embeddings as suggested in ForgetNet. In particular, inaccurate intermediate state predictions at the early stage of the training will potentially lead to sub-optimal convergence. To alleviate this, we propose the G-ForgetNet model, which uses a learnable gating mechanism and an additional regularization term in order to capture the Markov property of ForgetNet without the subsequent training limitations.

### 4.1 LIMITATIONS OF ENTIRELY REMOVING HISTORICAL EMBEDDINGS

While our ForgetNet model demonstrates effectiveness on a diverse set of tasks, it underperforms the baseline on several tasks, such as the Floyd-Warshall algorithm. A closer examination suggests that during the early stage of training, the model struggles with producing accurate intermediate predictions for certain algorithm tasks, which could lead the model towards suboptimal convergence. To make this clear, with a slight abuse of notations, we let $x$ and $y^{(t)}$ denote the input state and the $t$-th

intermediate state, *i.e.*, the `hints`, of an algorithmic trajectory sample, respectively. Accordingly, $\widehat{y}^{(t)}$ represents the intermediate state predicted at timestep $t$. In addition, $\mathcal{E}$, $\mathcal{P}$, and $\mathcal{D}$ represent the computation included in the encoder, processor, and decoder, respectively. In our ForgetNet model, the computational pathway $x \to \mathcal{E} \to \mathcal{P} \to \mathcal{D} \to y^{(1)}$ naturally emerges as a desirable pathway for training. This is because both $x$ and $y^{(1)}$ are accurate, facilitating a high-quality back-propagation signal. However, as we progress into extended paths for subsequent execution steps, we expose the model to the pitfalls of inaccurate intermediate state predictions. For example, the computational pathway associated with the loss function for the second intermediate state, $x \to \mathcal{E} \to \mathcal{P} \to \mathcal{D} \to \widehat{y}^{(1)} \to \mathcal{E} \to \mathcal{P} \to \mathcal{D} \to y^{(2)}$, is impacted by the inaccuracies of the prediction $\widehat{y}^{(1)}$. Intuitively, this introduces noise for the processor $\mathcal{P}$ to learn the one-step execution of the algorithm, since the processor receives inaccurate input at the second time step. Such inaccuracies accumulate over time steps. This indicates that, during the early stages of training, the model is primarily navigating these sub-optimal pathways, hindering its optimization. Additionally, by removing the hidden states in ForgetNet, we have essentially removed a residual connection between processor layers, making it more difficult for the model to backpropagate signals through many consecutive processor layers. As an empirical illustration, Figure 4 shows the training loss of ForgetNet and that of the baseline model for the Floyd-Warshall algorithm task. It indeed shows that the training losses for ForgetNet are elevated during the early stage of training, leading to sub-optimal convergence. We provide more results and deeper analysis of the training difficulties in ForgetNet in Appendix B.2, where we observe that elevated losses in ForgetNet are primarily incurred during the later steps of the `hints` time series, indicating the difficulties the model has with accumulation of inaccurate intermediate predictions.

## 4.2 G-FORGETNET: ADAPTIVE USE OF HISTORICAL EMBEDDINGS

In light of the aforementioned limitation, we further introduce G-ForgetNet with a regularized gating mechanism that restores important computational pathways during training and learns to align with the Markov property. The core motivation behind this proposal is that while inclusion of information from previous layers does not align with the inherent Markov nature of the task, it can provide helpful support, especially during the early stage of training, where it can mitigate the effects of inaccurate intermediate predictions and facilitate higher quality backpropagation signals. Further, the added loss penalty encourages the model to obey the Markov property that was shown to be beneficial in Section 3.2. Specifically, by including $\boldsymbol{h}_i^{(t-1)}$ in Eq. (2) as a component of the input for the processor at time step $t$, it can enrich the model with useful computational pathways, such as $x \to \mathcal{E} \to \mathcal{P} \to \mathcal{P} \to \mathcal{D} \to y^{(2)}$ associated with the loss function for the second intermediate state. In general, the introduced computational pathways $x \to \mathcal{E} \to \mathcal{P} \to \cdots \to \mathcal{P} \to \mathcal{D} \to y^{(t)}$, where there are $t$ sequentially applied processors, are valuable for training the processor $\mathcal{P}$ to capture one-step algorithmic execution. This is because $y^{(t)}$ is the accurate output after executing the algorithm for $t$ steps from the input $x$. In essence, these pathways create an alternative route, circumventing the challenges posed by inaccurate intermediate state predictions, especially at the early stage of training.

Based on the above intuition, in G-ForgetNet, we further introduce a learnable gating mechanism that modulates the use of historical embeddings. Formally, Eq. (2) is replaced with

$$\boldsymbol{z}_i^{(t)} = \left[ \bar{\boldsymbol{x}}_i^{(t)}, \boldsymbol{g}_i^{(t)} \odot \boldsymbol{h}_i^{(t-1)} \right], \quad \{\boldsymbol{h}_i^{(t)}\} = f_{\text{GNN}}\left( \{\boldsymbol{z}_i^{(t)}\}, \{\bar{\boldsymbol{e}}_{ij}^{(t)}\}, \bar{\boldsymbol{g}}^{(t)} \right), \tag{4}$$

where the gate $\boldsymbol{g}_i^{(t)}$ has the same dimensions as $\boldsymbol{h}_i^{(t-1)}$ and $\odot$ denotes element-wise product. Here, we employ a simple multi-layer perceptron (MLP) to obtain the gate as

$$\boldsymbol{g}_i^{(t)} = \sigma\left( \text{MLP}\left( \left[ \bar{\boldsymbol{x}}_i^{(t)}, \boldsymbol{h}_i^{(t-1)} \right] \right) \right), \tag{5}$$

where $\sigma(\cdot)$ is the sigmoid function. An illustration of G-ForgetNet is in Figure 1 (c). Finally, we introduce a modified `hints` loss function that includes a regularization term on the magnitude of $\boldsymbol{g}_i^{(t)}$ as

$$\text{Loss}^{(t)} = \mathcal{L}\left( \hat{y}^{(t)}, y^{(t)} \right) + \lambda \sum_i \left\| \boldsymbol{g}_i^{(t)} \right\| \tag{6}$$

Where $\mathcal{L}\left( \hat{y}^{(t)}, y^{(t)} \right)$ is the standard `hints` loss functions used in the CLRS-30 benchmark, which depends on the type and location of features contained in $y^{(t)}$. At the early stage of training, we

Table 1: Test OOD micro-F1 score of the baseline, ForgetNet, and G-ForgetNet methods. The cells are highlighted if their corresponding results are better than the baseline.

| Algorithm | Baseline | ForgetNet | G-ForgetNet | Algorithm | Baseline | ForgetNet | G-ForgetNet |
|---|---|---|---|---|---|---|---|
| Activity Selector | $93.02\%_{\pm1.62}$ | $97.17\%_{\pm0.20}$ | $99.03\%_{\pm0.10}$ | Jarvis' March | $85.44\%_{\pm3.26}$ | $85.21\%_{\pm2.83}$ | $88.53\%_{\pm2.96}$ |
| Articulation Points | $95.01\%_{\pm2.09}$ | $90.16\%_{\pm2.25}$ | $97.97\%_{\pm0.58}$ | Knuth-Morris-Pratt | $3.96\%_{\pm1.33}$ | $18.96\%_{\pm4.19}$ | $12.45\%_{\pm3.12}$ |
| Bellman-Ford | $97.67\%_{\pm0.28}$ | $98.45\%_{\pm0.13}$ | $99.18\%_{\pm0.11}$ | LCS Length | $76.24\%_{\pm1.38}$ | $84.60\%_{\pm0.47}$ | $85.43\%_{\pm0.47}$ |
| BFS | $99.45\%_{\pm0.11}$ | $99.32\%_{\pm0.16}$ | $99.96\%_{\pm0.01}$ | Matrix Chain Order | $86.31\%_{\pm0.86}$ | $94.83\%_{\pm0.43}$ | $91.08\%_{\pm0.51}$ |
| Binary Search | $62.79\%_{\pm4.31}$ | $74.41\%_{\pm2.11}$ | $85.96\%_{\pm1.59}$ | Minimum | $96.42\%_{\pm1.35}$ | $99.01\%_{\pm0.10}$ | $99.26\%_{\pm0.08}$ |
| Bridges | $89.58\%_{\pm4.79}$ | $94.74\%_{\pm2.00}$ | $99.43\%_{\pm0.15}$ | MST-Kruskal | $87.42\%_{\pm1.12}$ | $85.08\%_{\pm1.12}$ | $91.25\%_{\pm0.40}$ |
| Bubble Sort | $61.93\%_{\pm6.24}$ | $75.12\%_{\pm2.97}$ | $83.19\%_{\pm2.59}$ | MST-Prim | $92.35\%_{\pm0.87}$ | $94.14\%_{\pm0.50}$ | $95.19\%_{\pm0.33}$ |
| DAG Shortest Paths | $97.92\%_{\pm0.28}$ | $98.18\%_{\pm0.24}$ | $99.37\%_{\pm0.03}$ | Naïve String Matcher | $80.32\%_{\pm6.66}$ | $62.22\%_{\pm3.55}$ | $97.02\%_{\pm0.77}$ |
| DFS | $30.33\%_{\pm4.77}$ | $97.12\%_{\pm1.68}$ | $74.31\%_{\pm5.03}$ | Optimal BST | $78.14\%_{\pm1.26}$ | $80.19\%_{\pm0.76}$ | $83.58\%_{\pm0.49}$ |
| Dijkstra | $96.78\%_{\pm0.72}$ | $98.32\%_{\pm0.13}$ | $99.14\%_{\pm0.06}$ | Quickselect | $1.45\%_{\pm0.34}$ | $1.61\%_{\pm0.38}$ | $6.30\%_{\pm0.85}$ |
| Find Max. Subarray | $63.67\%_{\pm1.70}$ | $64.57\%_{\pm1.42}$ | $78.97\%_{\pm0.70}$ | Quicksort | $47.86\%_{\pm6.34}$ | $61.92\%_{\pm6.25}$ | $73.28\%_{\pm6.25}$ |
| Floyd-Warshall | $53.00\%_{\pm1.17}$ | $38.14\%_{\pm1.09}$ | $56.32\%_{\pm0.86}$ | Segments Intersect | $97.83\%_{\pm0.11}$ | $98.55\%_{\pm0.09}$ | $99.06\%_{\pm0.39}$ |
| Graham Scan | $91.82\%_{\pm1.20}$ | $95.49\%_{\pm0.27}$ | $97.67\%_{\pm0.14}$ | SCC | $44.83\%_{\pm2.74}$ | $50.80\%_{\pm2.67}$ | $53.53\%_{\pm2.48}$ |
| Heapsort | $48.09\%_{\pm5.42}$ | $46.90\%_{\pm5.96}$ | $57.47\%_{\pm6.08}$ | Task Scheduling | $80.93\%_{\pm0.21}$ | $86.31\%_{\pm0.46}$ | $84.55\%_{\pm0.35}$ |
| Insertion Sort | $70.62\%_{\pm8.26}$ | $95.19\%_{\pm0.77}$ | $98.40\%_{\pm0.21}$ | Topological Sort | $84.67\%_{\pm3.93}$ | $92.63\%_{\pm1.55}$ | $99.92\%_{\pm0.02}$ |

intuitively anticipate the gate to be more "open", *i.e.*, the magnitude of $\boldsymbol{g}_i^{(t)}$ to be large, thus enriching the model with the aforementioned beneficial pathways. As training progresses and the model starts predicting more reliable intermediate predictions, the dependence on historical embeddings should diminish, *i.e.*, the gate becomes more "closed", to honor the Markov nature. Since the scale of the `hints` losses varies drastically for each algorithm, we use a heuristic to select the value of $\lambda$ for each algorithm based on the loss values; further details can be found in Appendix A.1.

## 5 EXPERIMENTS

In this section, we perform comprehensive experiments to evaluate the proposed G-ForgetNet model, by addressing the following questions. (1) *Can our G-ForgetNet model, equipped with the regularized gating mechanism, consistently perform better than the baseline model? How does it perform in the several tasks where ForgetNet underperforms the baseline? Does it help the early stage of training?* (2) *How is the G-ForgetNet model compared to a boarder range of prior methods?* (3) *What are the dynamics of the gating mechanism within G-ForgetNet? To what extent does it align with our expectations?*

**Datasets and setup.** We perform experiments on the standard out-of-distribution (OOD) splits present in the CLRS-30 algorithmic reasoning benchmark (Veličković et al., 2022a). To be specific, we train on `inputs` with 16 or fewer nodes, and use `inputs` with 16 nodes for validation. During testing, for most algorithms, there are 32 trajectories with `inputs` of 64 nodes. For algorithms where the `ouputs` are associated with graph-level features, rather than node-level or edge-level, there are 64× more trajectories, ensuring a consistent number of targets across all tasks.

For the baseline, ForgetNet, and G-ForgetNet introduced in Section 3.1, 3.2, and 4.2 respectively, we conduct 10 runs for each model in each task, with a single set of hyperparameters. Specifically, we employ the Adam optimizer (Kingma & Ba, 2015) with a cosine learning rate scheduler and an initial learning rate of 0.0015. The models are trained for 10,000 steps with a batch size of 32.

**More baselines.** Beyond the baseline paradigm introduced in Section 3.1, we further include more existing state-of-the-art methods for comparison. Specifically, we first involve the notable methods studied in the CLRS-30 benchmark, including Memnet (Sukhbaatar et al., 2015), MPNN (Gilmer et al., 2017), and PGN (Veličković et al., 2020a). These models serve as the processor in the encoder-processor-decoder framework, which is trained with noisy teacher forcing in the benchmark setup. Furthermore, we include the recently proposed Triplet-GMPNN method (Ibarz et al., 2022), which develops a set of techniques to stabilize training, thus removing teacher forcing completely to align the training and inference. Moreover, the processor network in Triplet-GMPNN is a message passing network which incorporates messages from triplets of nodes. Our introduced baseline, ForgetNet, and G-ForgetNet, *i.e.*, the three methods in Figure 1, are built on the framework as developed by Ibarz et al. (2022). Differing from the baseline described in Section 3.1, Triplet-GMPNN has an additional update gate. It is worth noting that such a gate is different from our introduced gating mechanism in G-ForgetNet in terms of both motivation and architectural design. In particular, the update gate in Triplet-GMPNN is placed ahead of the decoder and aims to update the processed embeddings for a subset of nodes at each time step and keep the remaining unchanged, whereas our gate mechanism is

Table 2: Test OOD micro-F1 score of G-ForgetNet and existing methods. The highest scores are highlighted in bold, while the second-highest scores are underlined. Results for individual algorithms can be found in Table 3.

| Algorithm | Memnet | MPNN | PGN | Triplet-GMPNN | G-ForgetNet |
|---|---|---|---|---|---|
| Div. & C. | $13.05\%_{\pm0.08}$ | $20.30\%_{\pm0.49}$ | $65.23\%_{\pm2.56}$ | $\underline{76.36\%}_{\pm0.43}$ | $\mathbf{78.97\%}_{\pm0.70}$ |
| DP | $67.95\%_{\pm2.19}$ | $65.10\%_{\pm0.73}$ | $70.58\%_{\pm0.84}$ | $\underline{81.99\%}_{\pm1.30}$ | $\mathbf{86.70\%}_{\pm0.49}$ |
| Geometry | $45.14\%_{\pm2.36}$ | $73.11\%_{\pm4.27}$ | $61.19\%_{\pm1.14}$ | $\underline{94.09\%}_{\pm0.77}$ | $\mathbf{95.09\%}_{\pm1.16}$ |
| Graphs | $24.12\%_{\pm1.46}$ | $62.80\%_{\pm2.55}$ | $60.25\%_{\pm1.57}$ | $\underline{81.41\%}_{\pm1.53}$ | $\mathbf{88.80\%}_{\pm0.84}$ |
| Greedy | $53.42\%_{\pm1.13}$ | $82.39\%_{\pm1.74}$ | $75.85\%_{\pm1.27}$ | $\underline{91.22\%}_{\pm0.40}$ | $\mathbf{91.79\%}_{\pm0.23}$ |
| Search | $34.35\%_{\pm0.20}$ | $41.20\%_{\pm0.61}$ | $56.11\%_{\pm0.36}$ | $\underline{58.61\%}_{\pm1.05}$ | $\mathbf{63.84\%}_{\pm0.84}$ |
| Sorting | $\underline{71.53\%}_{\pm0.97}$ | $11.83\%_{\pm0.91}$ | $15.46\%_{\pm1.18}$ | $60.38\%_{\pm5.27}$ | $\mathbf{78.09\%}_{\pm3.78}$ |
| Strings | $1.52\%_{\pm0.24}$ | $3.21\%_{\pm0.58}$ | $2.04\%_{\pm0.16}$ | $\underline{49.09\%}_{\pm4.78}$ | $\mathbf{54.74\%}_{\pm1.95}$ |
| Overall Average | $38.03\%$ | $51.02\%$ | $52.31\%$ | $\underline{75.98\%}$ | $\mathbf{82.89\%}$ |
| > 99% | 0/30 | 1/30 | 1/30 | 1/30 | **9/30** |
| > 97% | 0/30 | 1/30 | 1/30 | 5/30 | **13/30** |
| > 95% | 0/30 | 2/30 | 2/30 | $\underline{7/30}$ | **14/30** |

designed to enforce the Markov property of algorithmic reasoning and is supported by a regularization term in the loss function. Another recent model, Hint-ReLIC (Bevilacqua et al., 2023), uses an additional self-supervised learning objective based on data augmentations. Given such augmentations and different setups, our model and Hint-ReLIC are not directly comparable. We expect that a fusion of our model and Hint-ReLIC could further boost the performance, and we leave such an evaluation to future work as the code of Hint-ReLIC is not yet publicly available.

**G-ForgetNet *vs*. ForgetNet *vs*. the baseline.** In Section 3.2, we have demonstrated the effectiveness of our ForgetNet model which removes historical embeddings to honor the Markov nature of algorithmic reasoning. Here, we further compare the three methods included in Figure 1 to evaluate the effectiveness of our proposed gating mechanism in the G-ForgetNet model. In Table 1, we report the average test results over 10 runs for each model in each algorithm. While ForgetNet surpasses the baseline across 23/30 tasks, G-ForgetNet consistently achieves improved performance over the baseline on all 30 tasks. In the several tasks where ForgetNet underperforms the baseline, such as the Floyd-Warshall and naïve string matcher tasks, G-ForgetNet demonstrates consistent improvements over the baseline. For example, in the naïve string matcher task, while ForgetNet performs worse than the baseine, G-ForgetNet outperforms the baseline by an absolute margin of $16.70\%$. This demonstrates the effectiveness of the proposed gating mechanism, which is able to capture the benefits of honoring the Markov property without the training difficulties of ForgetNet.

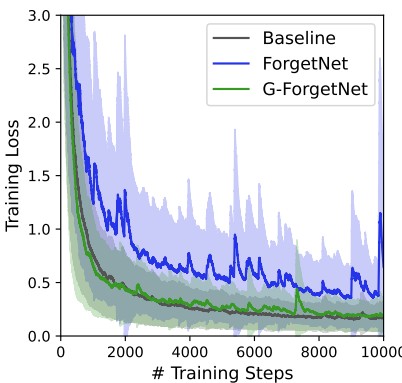

Figure 4: Training curves for the baseline, ForgetNet, and G-ForgetNet methods on the Floyd-Warshall task. The shaded region indicates the standard deviation. Figure is smoothed for clarity.

As clarified in Section 4.2, the proposed gating structure is expected to enhance the early stage of training, thus improving the final convergence in many tasks. To empirically verify such intuition, in Figure 4, we illustrate the training losses of the baseline, ForgetNet, and G-ForgetNet models in the Floyd-Warshall task. We observe that ForgetNet indeed faces challenges during the early stages, leading to sub-optimal convergence compared to the baseline in this task. The G-ForgetNet model, can effectively sidestep the early training pitfalls, thereby leading to a better convergence at the end of training in this task. This verifies our intuition that the additional computational pathways in G-ForgetNet can help enhance the early stages of training. In Appendix B we dive deeper into the loss curves corresponding to different execution steps for several algorithms and demonstrate that the loss experienced by ForgetNet at each execution step tends to escalate more sharply as the algorithmic execution progresses than G-ForgetNet. This observation validates our earlier intuition in Section 4.2 that the gating mechanism in G-ForgetNet introduces computational pathways that act as corrective

signals against accumulated errors. By offering these pathways, G-ForgetNet can circumvent the pitfalls posed by inaccurate intermediate predictions, thereby facilitating the optimization of the losses corresponding to later execution steps. Overall, G-ForgetNet outperforms ForgetNet in 26/30 tasks and improves the overall average score from 78.98% in ForgetNet to 82.89% in G-ForgetNet.

**Compared to more existing methods.** We further extend our comparison of G-ForgetNet to more existing methods, including the aforementioned Memnet, MPNN, PGN, and Triplet-GMPNN methods. The results of these methods are obtained from the respective literature (Veličković et al., 2022a; Ibarz et al., 2022). As summarized in Table 2, G-ForgetNet emerges as the top performer in 25/30 algorithmic tasks. Compared to the previous state-of-the-art method Triplet-GMPNN, G-ForgetNet improves the mean test performance across all 30 tasks from 75.98% to 82.89%. Additionally, G-ForgetNet surpasses the 99% threshold on 9/30 algorithms, compared to the prior best of just 1/30. Further, G-ForgetNet achieves large performance increases on several algorithms. For example, G-ForgetNet achieves a test score of 97.02% in the naïve string matcher task, while the previous state-of-the-art performance is 78.67%, and G-ForgetNet achieves a test score of 98.40% on insertion sort, compared to the previous state-of-the-art of 78.14%. This comparison further demonstrates the effectiveness of our proposed method.

**Dynamics of the gating mechanism.** In order to understand the behavior of the gating mechanism and gauge its alignment with our anticipations, we empirically investigate its dynamics during training. Specifically, we compute the L2 norm of the hidden states, $h_i^{(t)}$ before being passed to the processor and then normalize by dividing by the square root of the hidden state dimension. In G-ForgetNet, the L2 norm is taken after gating $g_i^{(t)} \odot h_i^{(t-1)}$, so we are effectively measuring how much of the hidden states are allowed to pass into the processor. For every sample in the validation set, we consistently monitor the average L2 norm over both nodes and algorithmic execution steps, along the training iterations. In Figure 5, we illustrate the average L2 norm over all samples in the validation set during the training process for the Floyd-Warshall task for the baseline and for G-ForgetNet. We observe that the baseline hidden state norm is fairly constant and has a relatively large magnitude, indicating that

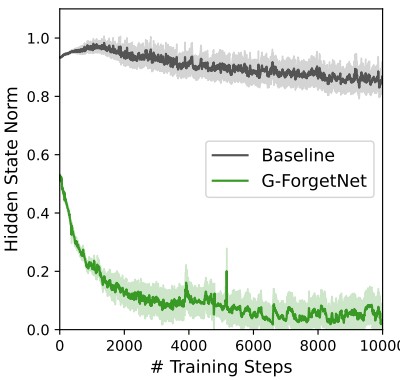

Figure 5: Average L2 norm value throughout the training process on the Floyd-Warshall task. The shaded region indicates the standard deviation.

it is fitting historical noise during training, whereas G-ForgetNet declines to nearly zero. This empirically validates that the dynamics of the gating mechanism align with our intuition in this task. That is, the gating mechanism is open during the beginning of training, thus enhancing early training while progressively focusing on the Markov nature of algorithmic tasks. We generally observe similar trends across all of the CLRS-30 algorithms, with more tasks shown in Appendix A.2. We further validate the importance of the loss penalty included in G-ForgetNet in Appendix A.3, where we investigate the behavior of the G-ForgetNet model without the loss penalty. We observe that without the loss penalty, the model still exhibits declining trends in the hidden state norm, however it will not converge to 0 as desired. The performance of G-ForgetNet without the penalty is still better than the baselines, however the performance is significantly improved with the penalty. This aligns with our intuitions since the penalty ensures that G-ForgetNet is consistent with the Markov property.

## 6 CONCLUSION

In this work, we highlight a key misalignment between the prevalent practice of incorporating historical embeddings and the intrinsic Markov characteristics of algorithmic reasoning tasks. In response, we propose ForgetNet, which explicitly honors the Markov nature by removing the use of historical embeddings, and its adaptive variant, G-ForgetNet, equipped with a gating mechanism and subsequent loss penalty in order to capture the benefits of the Markov property without the training difficulties found in ForgetNet. Our comprehensive experiments on the CLRS-30 benchmark demonstrate the superior generalization capabilities of both models compared to established baselines. In summary, this work reveals the importance of aligning model design with the Markov nature in neural algorithmic reasoning tasks, paving the way for more advancements in future research.

ACKNOWLEDGMENTS

This work was supported in part by National Science Foundation grants IIS-2243850 and IIS-2006861.

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

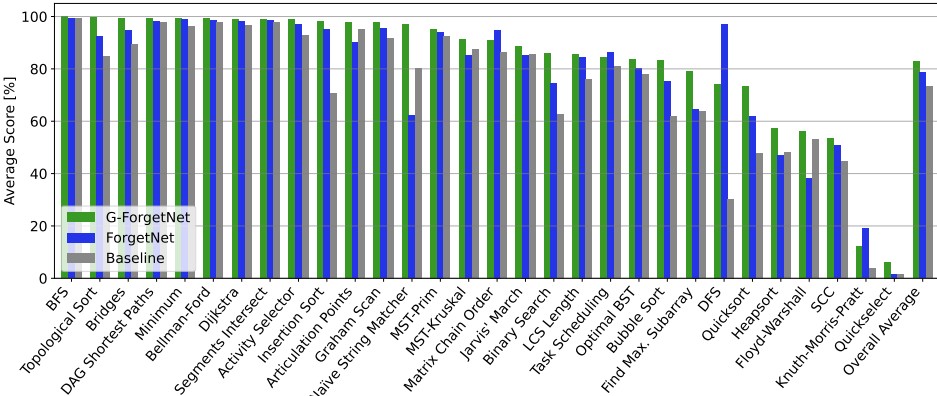

Figure 6: Comparison between the baseline, ForgetNet, and G-ForgetNet. Reported results are the average of 10 runs with random seeds. Numerical results can be found in Table1

## A    FURTHER G-FORGETNET ANALYSIS

In Figure 6 we compare G-ForgetNet with ForgetNet and the baseline model, and in Table 3, we provide the full numeric results for G-ForgetNet compared with Memnet, MPNN, PGN, and Triplet-GMPNN.

### A.1    LAMBDA HEURISTIC

Since the scale of the loss function varies drastically between algorithms, it is not possible to use a single value for $\lambda$ across all algorithms. In general, it is non-trivial to select optimal values for this parameter, and in this work, we use a heuristic to select reasonable values for $\lambda$. Specifically, we select $\lambda$ such that the gate penalty makes up approximately half of the total training loss after 6,000 training steps. Intuitively, with such a schedule, the model will spend the first 6,000 steps simply learning to execute the algorithm, then during the remaining 4,000 steps, the model will focus on learning single-step executions in accordance with the Markov property. As demonstrated in Table 2, this simple heuristic has quite robust performance across the entire set of CLRS-30 algorithms. We acknowledge that such a simple penalty schedule and heuristic is unlikely to be optimal and will be approved upon by future works.

### A.2    GATE ANALYSIS

In Figure 7 we include more figures of the hidden state's L2 norm for G-ForgetNet and the baseline, as in Section 5. These further support that our G-ForgetNet model does enforce the Markov property during testing as we observe the L2 norm converge to 0.

Table 3: Test OOD micro-F1 score of G-ForgetNet and existing methods. The highest scores are highlighted in bold, while the second-highest scores are underlined.

| Algorithm | Memnet | MPNN | PGN | Triplet-GMPNN | G-ForgetNet |
|---|---|---|---|---|---|
| Activity Selector | $24.10\%_{\pm 2.22}$ | $80.66\%_{\pm 3.16}$ | $66.80\%_{\pm 1.62}$ | $\underline{95.18\%}_{\pm 0.45}$ | $\mathbf{99.03\%}_{\pm 0.10}$ |
| Articulation Points | $1.50\%_{\pm 0.61}$ | $50.91\%_{\pm 2.18}$ | $49.53\%_{\pm 2.09}$ | $\underline{88.32\%}_{\pm 2.01}$ | $\mathbf{97.97\%}_{\pm 0.46}$ |
| Bellman-Ford | $40.04\%_{\pm 1.46}$ | $92.01\%_{\pm 0.28}$ | $92.99\%_{\pm 0.34}$ | $\underline{97.39\%}_{\pm 0.19}$ | $\mathbf{99.18\%}_{\pm 0.11}$ |
| BFS | $43.34\%_{\pm 0.04}$ | $\underline{99.89\%}_{\pm 0.05}$ | $99.63\%_{\pm 0.29}$ | $99.73\%_{\pm 0.04}$ | $\mathbf{99.96\%}_{\pm 0.01}$ |
| Binary Search | $14.37\%_{\pm 0.46}$ | $36.83\%_{\pm 0.26}$ | $76.95\%_{\pm 0.13}$ | $\underline{77.58\%}_{\pm 2.35}$ | $\mathbf{85.96\%}_{\pm 1.59}$ |
| Bridges | $30.26\%_{\pm 0.05}$ | $72.69\%_{\pm 4.78}$ | $51.42\%_{\pm 7.82}$ | $\underline{93.99\%}_{\pm 2.07}$ | $\mathbf{99.43\%}_{\pm 0.15}$ |
| Bubble Sort | $\underline{73.58\%}_{\pm 0.78}$ | $5.27\%_{\pm 0.60}$ | $6.01\%_{\pm 1.95}$ | $67.68\%_{\pm 5.50}$ | $\mathbf{83.19\%}_{\pm 2.59}$ |
| DAG Shortest Paths | $66.15\%_{\pm 1.92}$ | $96.24\%_{\pm 0.56}$ | $96.94\%_{\pm 0.16}$ | $\underline{98.19\%}_{\pm 0.30}$ | $\mathbf{99.37\%}_{\pm 0.03}$ |
| DFS | $13.36\%_{\pm 1.61}$ | $6.54\%_{\pm 0.51}$ | $8.71\%_{\pm 0.24}$ | $\underline{47.79\%}_{\pm 4.19}$ | $\mathbf{74.31\%}_{\pm 5.03}$ |
| Dijkstra | $22.48\%_{\pm 2.39}$ | $91.50\%_{\pm 0.50}$ | $83.45\%_{\pm 1.75}$ | $\underline{96.05\%}_{\pm 0.60}$ | $\mathbf{99.14\%}_{\pm 0.06}$ |
| Find Max. Subarray | $13.05\%_{\pm 0.08}$ | $20.30\%_{\pm 0.49}$ | $65.23\%_{\pm 2.56}$ | $\underline{76.36\%}_{\pm 0.43}$ | $\mathbf{78.97\%}_{\pm 0.70}$ |
| Floyd-Warshall | $14.17\%_{\pm 0.13}$ | $26.74\%_{\pm 1.77}$ | $28.76\%_{\pm 0.51}$ | $\underline{48.52\%}_{\pm 1.04}$ | $\mathbf{56.32\%}_{\pm 0.86}$ |
| Graham Scan | $40.62\%_{\pm 2.31}$ | $91.04\%_{\pm 0.31}$ | $56.87\%_{\pm 1.61}$ | $\underline{93.62\%}_{\pm 0.91}$ | $\mathbf{97.67\%}_{\pm 0.14}$ |
| Heapsort | $\mathbf{68.00\%}_{\pm 1.57}$ | $10.94\%_{\pm 0.84}$ | $5.27\%_{\pm 0.18}$ | $31.04\%_{\pm 5.82}$ | $\underline{57.47\%}_{\pm 6.08}$ |
| Insertion Sort | $71.42\%_{\pm 0.86}$ | $19.81\%_{\pm 2.08}$ | $44.37\%_{\pm 2.43}$ | $\underline{78.14\%}_{\pm 4.64}$ | $\mathbf{98.40\%}_{\pm 0.21}$ |
| Jarvis' March | $22.99\%_{\pm 3.87}$ | $34.86\%_{\pm 12.39}$ | $49.19\%_{\pm 1.07}$ | $\mathbf{91.01\%}_{\pm 1.30}$ | $\underline{88.53\%}_{\pm 2.96}$ |
| Knuth-Morris-Pratt | $1.81\%_{\pm 0.00}$ | $2.49\%_{\pm 0.86}$ | $2.00\%_{\pm 0.12}$ | $\mathbf{19.51\%}_{\pm 4.57}$ | $\underline{12.45\%}_{\pm 3.12}$ |
| LCS Length | $49.84\%_{\pm 4.34}$ | $53.23\%_{\pm 0.36}$ | $56.82\%_{\pm 0.21}$ | $\underline{80.51\%}_{\pm 1.84}$ | $\mathbf{85.43\%}_{\pm 0.47}$ |
| Matrix Chain Order | $81.96\%_{\pm 1.03}$ | $79.84\%_{\pm 1.40}$ | $83.91\%_{\pm 0.49}$ | $\mathbf{91.68\%}_{\pm 0.59}$ | $\underline{91.08\%}_{\pm 0.51}$ |
| Minimum | $86.93\%_{\pm 0.11}$ | $85.34\%_{\pm 0.88}$ | $87.71\%_{\pm 0.52}$ | $\underline{97.78\%}_{\pm 0.55}$ | $\mathbf{99.26\%}_{\pm 0.08}$ |
| MST-Kruskal | $28.84\%_{\pm 0.61}$ | $70.97\%_{\pm 1.50}$ | $66.96\%_{\pm 1.36}$ | $\underline{89.80\%}_{\pm 0.77}$ | $\mathbf{91.25\%}_{\pm 0.40}$ |
| MST-Prim | $10.29\%_{\pm 3.77}$ | $69.08\%_{\pm 7.56}$ | $63.33\%_{\pm 0.98}$ | $\underline{86.39\%}_{\pm 1.33}$ | $\mathbf{95.19\%}_{\pm 0.33}$ |
| Naïve String Matcher | $1.22\%_{\pm 0.48}$ | $3.92\%_{\pm 0.30}$ | $2.08\%_{\pm 0.20}$ | $\underline{78.67\%}_{\pm 4.99}$ | $\mathbf{97.02\%}_{\pm 0.77}$ |
| Optimal BST | $72.03\%_{\pm 1.21}$ | $62.23\%_{\pm 0.44}$ | $71.01\%_{\pm 1.82}$ | $\underline{73.77\%}_{\pm 1.48}$ | $\mathbf{83.58\%}_{\pm 0.49}$ |
| Quickselect | $1.74\%_{\pm 0.03}$ | $1.43\%_{\pm 0.69}$ | $\underline{3.66\%}_{\pm 0.42}$ | $0.47\%_{\pm 0.25}$ | $\mathbf{6.30\%}_{\pm 0.85}$ |
| Quicksort | $\underline{73.10\%}_{\pm 0.67}$ | $11.30\%_{\pm 0.10}$ | $6.17\%_{\pm 0.15}$ | $64.64\%_{\pm 5.12}$ | $\mathbf{73.28\%}_{\pm 6.25}$ |
| Segments Intersect | $71.80\%_{\pm 0.90}$ | $93.44\%_{\pm 0.10}$ | $77.51\%_{\pm 0.75}$ | $\underline{97.64\%}_{\pm 0.09}$ | $\mathbf{99.06\%}_{\pm 0.39}$ |
| SCC | $16.32\%_{\pm 4.78}$ | $24.37\%_{\pm 4.88}$ | $20.80\%_{\pm 0.64}$ | $\underline{43.43\%}_{\pm 3.15}$ | $\mathbf{53.53\%}_{\pm 2.48}$ |
| Task Scheduling | $82.74\%_{\pm 0.04}$ | $84.11\%_{\pm 0.32}$ | $\underline{84.89\%}_{\pm 0.91}$ | $\mathbf{87.25\%}_{\pm 0.35}$ | $84.55\%_{\pm 0.35}$ |
| Topological Sort | $2.73\%_{\pm 0.11}$ | $52.60\%_{\pm 6.24}$ | $60.45\%_{\pm 2.69}$ | $\underline{87.27\%}_{\pm 2.67}$ | $\mathbf{99.92\%}_{\pm 0.02}$ |
| Overall Average | $38.03\%$ | $51.02\%$ | $52.31\%$ | $\underline{75.98\%}$ | $\mathbf{82.89\%}$ |
| $> 99\%$ | 0/30 | 1/30 | 1/30 | 1/30 | **9/30** |
| $> 97\%$ | 0/30 | 1/30 | 1/30 | $\underline{5/30}$ | **13/30** |
| $> 95\%$ | 0/30 | 2/30 | 2/30 | $\underline{7/30}$ | **14/30** |

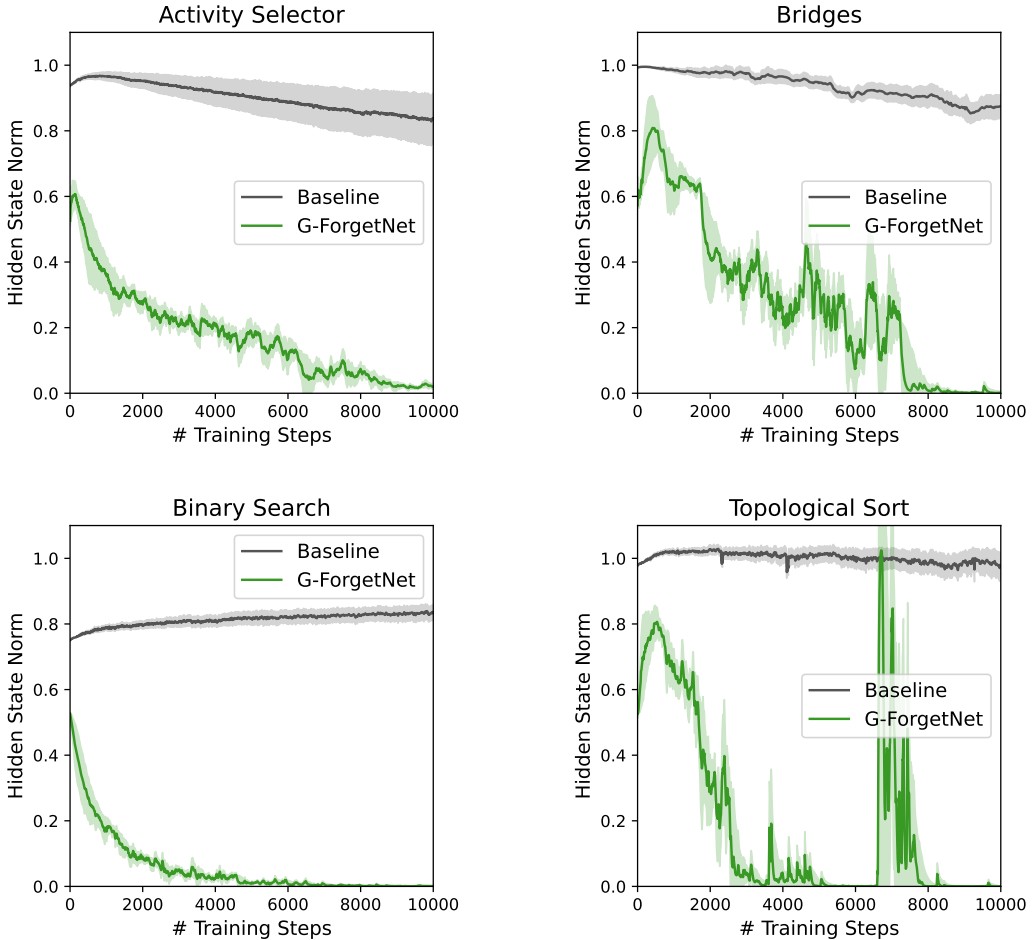

Figure 7: Average L2 norm value throughout the training process on the activity selector, bridges, binary search, and topological sort algorithms. Shaded regions indicate standard deviation.

Table 4: Test OOD micro-F1 score of the baseline, G-ForgetNet without penalty, and G-ForgetNet methods. The cells are highlighted if their corresponding results are better than the baseline. The highest scores are highlighted in bold.

| Algorithm | Baseline | G-ForgetNet w/o penalty | G-ForgetNet | Algorithm | Baseline | G-ForgetNet w/o penalty | G-ForgetNet |
|---|---|---|---|---|---|---|---|
| Activity Selector | $93.02\%_{\pm1.62}$ | $96.76\%_{\pm0.34}$ | $\mathbf{99.03\%}_{\pm0.10}$ | Jarvis' March | $85.44\%_{\pm3.26}$ | $\mathbf{88.61\%}_{\pm2.58}$ | $88.53\%_{\pm2.96}$ |
| Articulation Points | $95.01\%_{\pm2.09}$ | $\mathbf{98.97\%}_{\pm0.15}$ | $97.97\%_{\pm0.58}$ | Knuth-Morris-Pratt | $3.96\%_{\pm1.33}$ | $3.84\%_{\pm0.79}$ | $\mathbf{12.45\%}_{\pm3.12}$ |
| Bellman-Ford | $97.67\%_{\pm0.28}$ | $98.85\%_{\pm0.14}$ | $\mathbf{99.18\%}_{\pm0.11}$ | LCS Length | $76.24\%_{\pm1.38}$ | $80.33\%_{\pm1.68}$ | $\mathbf{85.43\%}_{\pm0.47}$ |
| BFS | $99.45\%_{\pm0.11}$ | $99.57\%_{\pm0.09}$ | $\mathbf{99.96\%}_{\pm0.01}$ | Matrix Chain Order | $86.31\%_{\pm0.86}$ | $86.83\%_{\pm1.23}$ | $\mathbf{91.08\%}_{\pm0.51}$ |
| Binary Search | $62.79\%_{\pm4.31}$ | $82.49\%_{\pm1.66}$ | $\mathbf{85.96\%}_{\pm1.59}$ | Minimum | $96.42\%_{\pm1.35}$ | $\mathbf{99.56\%}_{\pm0.03}$ | $99.26\%_{\pm0.08}$ |
| Bridges | $89.58\%_{\pm4.79}$ | $96.38\%_{\pm1.46}$ | $\mathbf{99.43\%}_{\pm0.15}$ | MST-Kruskal | $87.42\%_{\pm1.12}$ | $\mathbf{91.38\%}_{\pm0.58}$ | $91.25\%_{\pm0.40}$ |
| Bubble Sort | $61.93\%_{\pm6.24}$ | $64.78\%_{\pm3.71}$ | $\mathbf{83.19\%}_{\pm2.59}$ | MST-Prim | $92.35\%_{\pm0.87}$ | $94.51\%_{\pm0.88}$ | $\mathbf{95.19\%}_{\pm0.33}$ |
| DAG Shortest Paths | $97.92\%_{\pm0.28}$ | $98.70\%_{\pm0.20}$ | $\mathbf{99.37\%}_{\pm0.03}$ | Naïve String Matcher | $80.32\%_{\pm6.66}$ | $93.79\%_{\pm1.63}$ | $\mathbf{97.02\%}_{\pm0.77}$ |
| DFS | $30.33\%_{\pm4.77}$ | $47.97\%_{\pm4.42}$ | $\mathbf{74.31\%}_{\pm5.03}$ | Optimal BST | $78.14\%_{\pm1.26}$ | $79.14\%_{\pm1.73}$ | $\mathbf{83.58\%}_{\pm0.49}$ |
| Dijkstra | $96.78\%_{\pm0.72}$ | $98.46\%_{\pm0.23}$ | $\mathbf{99.14\%}_{\pm0.06}$ | Quickselect | $1.45\%_{\pm0.34}$ | $2.06\%_{\pm0.52}$ | $\mathbf{6.30\%}_{\pm0.85}$ |
| Find Max. Subarray | $63.67\%_{\pm1.70}$ | $77.65\%_{\pm1.05}$ | $\mathbf{78.97\%}_{\pm0.70}$ | Quicksort | $47.86\%_{\pm6.34}$ | $70.17\%_{\pm3.98}$ | $\mathbf{73.28\%}_{\pm6.25}$ |
| Floyd-Warshall | $53.00\%_{\pm1.17}$ | $54.60\%_{\pm1.14}$ | $\mathbf{56.32\%}_{\pm0.86}$ | Segments Intersect | $97.83\%_{\pm0.11}$ | $\mathbf{99.27\%}_{\pm0.05}$ | $99.06\%_{\pm0.39}$ |
| Graham Scan | $91.82\%_{\pm1.20}$ | $97.32\%_{\pm0.26}$ | $\mathbf{97.67\%}_{\pm0.14}$ | SCC | $44.83\%_{\pm2.74}$ | $59.78\%_{\pm2.80}$ | $53.53\%_{\pm2.48}$ |
| Heapsort | $48.09\%_{\pm5.42}$ | $\mathbf{59.09\%}_{\pm8.80}$ | $57.47\%_{\pm6.08}$ | Task Scheduling | $80.93\%_{\pm0.21}$ | $80.28\%_{\pm0.22}$ | $\mathbf{84.55\%}_{\pm0.35}$ |
| Insertion Sort | $70.62\%_{\pm8.26}$ | $87.90\%_{\pm2.59}$ | $\mathbf{98.40\%}_{\pm0.21}$ | Topological Sort | $84.67\%_{\pm3.93}$ | $90.36\%_{\pm2.84}$ | $\mathbf{99.92\%}_{\pm0.02}$ |

### A.3 PENALTY ANALYSIS

In Table 4, we report the performance of our G-ForgetNet model without the loss penalty, i.e., with just the gate mechanism. We observe that, even without the loss penalty, the model still outperforms the baseline on 28/30 algorithms, however the average score is only 79.31% compared to 82.88% with the penalty. Additionally, G-ForgetNet without penalty outperforms G-ForgetNet with penalty on 7 algorithms, however these cases are very small improvements. Overall, this study empirically shows us the importance of the penalty in the G-ForgetNet model, which aligns with our intuition that the penalty is necessary in order to enforce the Markov property. Finally, we provide a comparison of the L2 norm of the gated hidden states in G-ForgetNet with and without the penalty in Figure 8. On the activity selector algorithm, the G-ForgetNet model without the penalty still consistently decreases during training, however it does not reach the same final convergence as the model with the penalty, and on Floyd-Warshall, G-ForgetNet without the penalty is fairly constant throughout training, again demonstrating the necessity of the loss penalty to enforce the Markov property in G-ForgetNet.

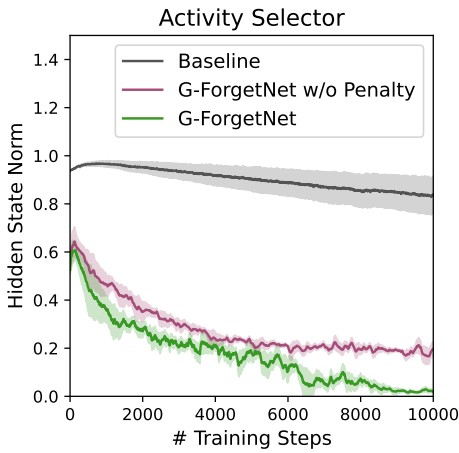 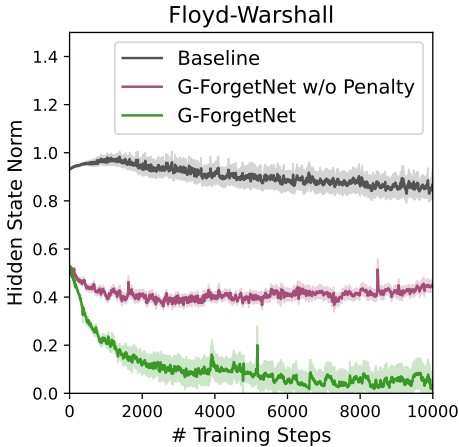

Figure 8: Average hidden state L2 norm value throughout the training process on the activity selector and Floyd-Warshall algorithms. Shaded regions indicate standard deviation.

## B MORE EXPERIMENTAL RESULTS

### B.1 MULTI-TASK EXPERIMENTS

Prior works (Xhonneux et al., 2021; Ibarz et al., 2022) have investigated jointly learning multiple algorithms using a single processor. We follow the multi-task setup in Ibarz et al. (2022) and train a single ForgetNet processor on all 30 CLRS algorithms while keeping separate encoders and decoders

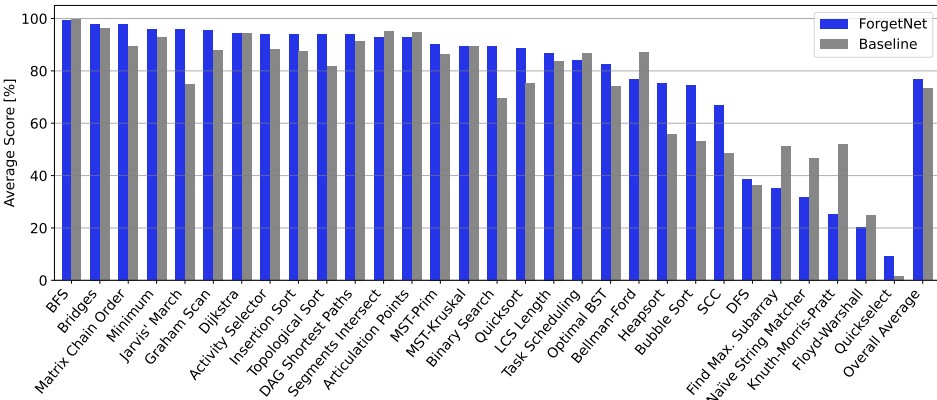

Figure 9: Comparison between the ForgetNet multi-task model and the baseline multi-task model. Average of 10 runs with random seeds. Numerical results can be found in Table 5.

Table 5: Test OOD micro-F1 scores of the baseline and ForgetNet on the multi-task setting.

| Algorithm | Baseline | ForgetNet | Algorithm | Baseline | ForgetNet |
|---|---|---|---|---|---|
| Activity Selector | $88.14\%_{\pm0.69}$ | $\mathbf{94.16}\%_{\pm0.55}$ | Jarvis' March | $74.84\%_{\pm5.34}$ | $\mathbf{95.79}\%_{\pm0.20}$ |
| Articulation Points | $\mathbf{94.70}\%_{\pm0.55}$ | $92.69\%_{\pm0.69}$ | Knuth-Morris-Pratt | $\mathbf{51.93}\%_{\pm4.33}$ | $25.16\%_{\pm0.85}$ |
| Bellman-Ford | $\mathbf{87.21}\%_{\pm1.56}$ | $76.41\%_{\pm3.45}$ | LCS Length | $83.68\%_{\pm0.57}$ | $\mathbf{86.85}\%_{\pm0.16}$ |
| BFS | $\mathbf{99.77}\%_{\pm0.05}$ | $99.32\%_{\pm0.29}$ | Matrix Chain Order | $89.60\%_{\pm0.49}$ | $\mathbf{97.83}\%_{\pm0.13}$ |
| Binary Search | $69.63\%_{\pm2.85}$ | $\mathbf{89.52}\%_{\pm1.23}$ | Minimum | $92.73\%_{\pm1.24}$ | $\mathbf{95.87}\%_{\pm0.50}$ |
| Bridges | $96.38\%_{\pm1.02}$ | $\mathbf{98.00}\%_{\pm0.98}$ | MST-Kruskal | $89.30\%_{\pm0.81}$ | $\mathbf{89.56}\%_{\pm1.68}$ |
| Bubble Sort | $53.24\%_{\pm5.01}$ | $\mathbf{74.40}\%_{\pm3.09}$ | MST-Prim | $86.54\%_{\pm0.93}$ | $\mathbf{90.38}\%_{\pm0.21}$ |
| DAG Shortest Paths | $91.49\%_{\pm0.76}$ | $\mathbf{93.87}\%_{\pm0.36}$ | Naïve String Matcher | $\mathbf{46.74}\%_{\pm3.51}$ | $31.90\%_{\pm2.00}$ |
| DFS | $36.35\%_{\pm2.99}$ | $\mathbf{38.51}\%_{\pm1.48}$ | Optimal BST | $74.23\%_{\pm2.18}$ | $\mathbf{82.55}\%_{\pm0.59}$ |
| Dijkstra | $\mathbf{94.49}\%_{\pm0.47}$ | $94.34\%_{\pm0.40}$ | Quickselect | $1.71\%_{\pm0.49}$ | $\mathbf{9.15}\%_{\pm0.57}$ |
| Find Max. Subarray | $\mathbf{51.15}\%_{\pm1.23}$ | $35.19\%_{\pm0.61}$ | Quicksort | $75.31\%_{\pm4.68}$ | $\mathbf{88.48}\%_{\pm2.13}$ |
| Floyd-Warshall | $\mathbf{24.99}\%_{\pm1.42}$ | $20.39\%_{\pm0.40}$ | Segments Intersect | $\mathbf{95.26}\%_{\pm0.25}$ | $92.71\%_{\pm0.35}$ |
| Graham Scan | $88.08\%_{\pm1.90}$ | $\mathbf{95.42}\%_{\pm0.25}$ | SCC | $48.73\%_{\pm4.98}$ | $\mathbf{66.93}\%_{\pm2.95}$ |
| Heapsort | $55.83\%_{\pm7.91}$ | $\mathbf{78.39}\%_{\pm4.92}$ | Task Scheduling | $\mathbf{86.61}\%_{\pm0.21}$ | $84.02\%_{\pm0.13}$ |
| Insertion Sort | $87.48\%_{\pm2.03}$ | $\mathbf{93.98}\%_{\pm0.92}$ | Topological Sort | $81.85\%_{\pm1.21}$ | $\mathbf{93.88}\%_{\pm0.48}$ |

for each algorithm. As shown in Figure 9 and in Table 5, ForgetNet performs better than the baseline on 20/30 algorithms and has a higher average score across all algorithms. This confirms that enforcing the Markov property is also beneficial in the multi-task setting.

## B.2 MORE TRAINING CURVES

**More training curves of the baseline, ForgetNet, and G-ForgetNet methods.** In Figure 10, we illustrate the training curves, including the total loss and losses at different execution steps, of the three methods in the Floyd-Warshall and activity selector tasks. We observe that ForgetNet does not converge to the same total loss value, shown in plot (a) on each figure. Further, we observe that the gap between ForgetNet and the baseline widens at later points in the `hints` time series, showing how the removal of connections between consecutive layers in ForgetNet introduces training difficulties. The inaccurate intermediate predictions cause the model to struggle to optimize losses corresponding to later execution steps.

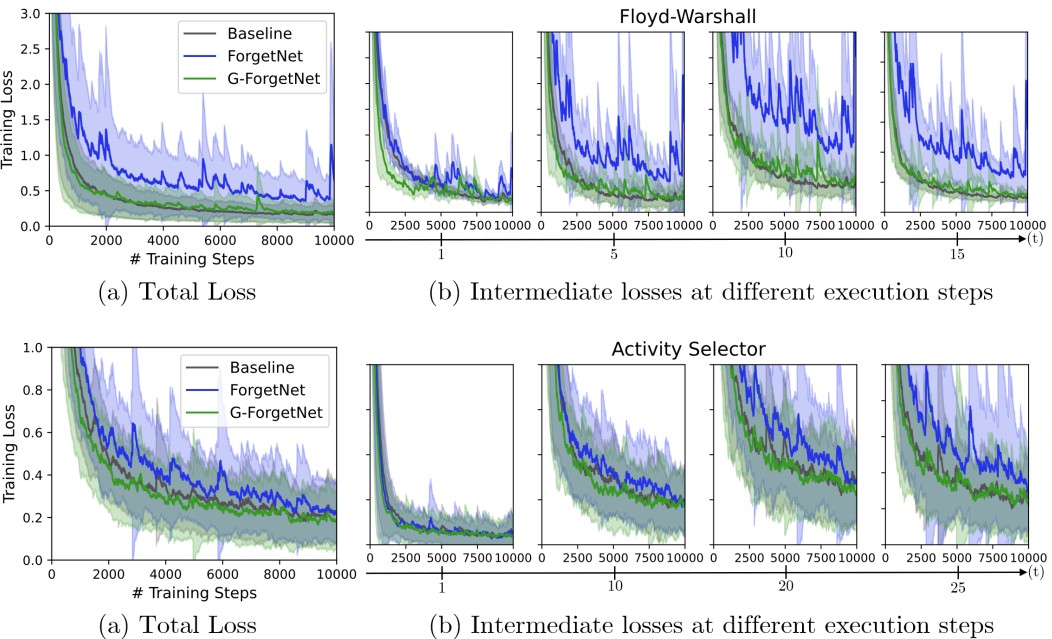

Figure 10: Training curves for the baseline, ForgetNet, and G-ForgetNet methods in the Floyd-Warshall and activity selector tasks, tasks (from top to bottom): (a) total loss and (b) losses at different execution steps, i.e. the losses incurred at different points in the `hints` time series.

