# OpenReview forum: "On the Markov Property of Neural Algorithmic Reasoning: Analyses and Methods"
_ICLR.cc/2024/Conference — ICLR 2024 spotlight_

### Official Review · Reviewer_QaJ5 · 2023-10-21

**Soundness:** 3 good
**Presentation:** 4 excellent
**Contribution:** 3 good
**Rating:** 6
**Confidence:** 3

**Summary:**

The paper points out that most algorithmic tasks should have Markov property and thus the optimal model should not rely on historical embeddings. The paper then proposes ForgeNet and GForgerNet which doesn't use or only limitedly use historical embeddings.

**Strengths:**

1. There is a novelty in observing the Markov property of many algorithmic tasks and proposing an architecture that aligns with this property.
2. The overall performance seems to be better than all compared baselines.
3. The paper is quite well-written and easy to follow.

**Weaknesses:**

The authors claimed that the gating value should decrease because  "As training progresses and the model starts predicting more reliable intermediate predictions". But if there are no regularisation terms on the gating variable, why should the gating value decrease if there is still useful information in the historical embeddings?  The authors do show a plot of the gating variable trajectory as training progresses, but the gating value only decreases slightly and never to 0. Are there any cases where the gating variable decreases to 0? If not, why is this the case since the model can just simply ignore the historical embedding after training stabilizes?

**Questions:**

See weakness,

---

> ### Author Response · Authors · 2023-11-20
> **Reponse to Reviewer QaJ5**
>
> Dear reviewer QaJ5, thank you for your valuable comments. Below are our responses to your concerns.
>
> > If there are no regularization terms on the gating variable, why should the gating value decrease if there is still useful information in the historical embeddings?
>
> Motivated by this comment, we have added a regularization term to G-ForgetNet and demonstrated its superb effectiveness for enforcing the Markov property in G-ForgetNet and improving model performance. We refer the reviewer to Section 4.2 and Section 5 to see the details of this new regularization term as well as an analysis of its performance and behavior. However, without this penalty term, we still observe that the gating value in G-ForgetNet decreases. As motivated by the Markov property and empirically shown in ForgetNet, the historical embeddings do not contain useful information, which is why we still observe that the gating value in G-ForgetNet decrease, allthough not to 0 as it does with the added penalty.
>
> > The authors do show a plot of the gating variable trajectory as training progresses, but the gating value only decreases slightly and never to 0. Are there any cases where the gating variable decreases to 0?
>
> Without the added loss penalty, the gating value generally will not converge exactly to 0, allthough it does get close for many algorithms. However, when we add the penalty to G-ForgetNet, we see that the gating value *does* converge to 0. We analyze the behavior of the penalized gate in Section 5 and in Appendix A.2, and we have included an in-depth comparison of G-ForgetNet with and without this penalty in Appendix A.3 in order to address these questions.
>
> Finally, we would like to thank the reviewer for the insightful points they raised

---

### Official Review · Reviewer_gWrY · 2023-10-30

**Soundness:** 3 good
**Presentation:** 4 excellent
**Contribution:** 3 good
**Rating:** 8
**Confidence:** 2

**Summary:**

This paper aims to reconsider the Markov property of neural algorithmic reasoning. While classical algorithms inherently possess the Markov property, existing neural algorithmic reasoning approaches learn to execute algorithms using historical embeddings, which contradicts this property. Building on this observation, this paper proposes ForgetNet, a method that eliminates the reliance on historical embeddings to reintroduce the Markov nature. Additionally, the paper introduces G-ForgetNet, which adaptively integrates historical embeddings.

**Strengths:**

1. This paper focuses on an emerging and interesting research topic: neural algorithmic reasoning.
2. This paper is well-written. Figures and formulas clearly illustrate the background and proposed method.
3. The motivation for aligning current algorithms with their Markov nature is well stated. Additionally, this paper complements a crucial concept in the context of neural algorithmic reasoning.
4. Experiments show that the proposed method can outperform baselines in most of the tasks (algorithms).

**Weaknesses:**

1. Although ForgetNet perfectly possesses the Markov property of algorithm execution, the newly proposed G-ForgetNet still utilizes historical embeddings but outperforms ForgetNet. Is there an inherent drawback of neural algorithmic reasoning in possessing the Markov property?
2. For certain tasks, G-ForgetNet performs worse than ForgetNet, or ForgetNet performs worse than the baseline. Some task-specific explanations regarding this should be discussed.

**Questions:**

Please see the weaknesses above.

---

> ### Author Response · Authors · 2023-11-20
> **Response to Reviewer gWrY**
>
> Dear reviewer gWrY, thank you for your valuable comments. Below are our responses to your concerns.
>
> > W1.  Although ForgetNet perfectly possesses the Markov property of algorithm execution, the newly proposed G-ForgetNet still utilizes historical embeddings but outperforms ForgetNet. Is there an inherent drawback of neural algorithmic reasoning in possessing the Markov property?
>
> The drawback of ForgetNet is not due to the Markov property, in fact, the Markov property is ForgetNet's biggest strength. The drawback of ForgetNet is the training difficulties that stem from removing the connections between layers. We have tried to clarify this point in Sections 4.1 and 4.2 of our revised manuscript. Additionally, motivated by comments from reviewers, we added a loss penalty to G-ForgetNet which better aligns G-ForgetNet with the Markov property. We demonstrate that, with the new loss penalty, the gate norm goes to 0 during training, *i.e.,* the gate becomes entirely closed. We observe that enforcing the gate to be closed significantly improves the performance of G-ForgetNet, as it now outperforms the baseline on all 30/30 algorithms. As such, we further show the power of aligning neural algorithmic reasoners with the important Markov property. We refer the reviewer to the revised Section 5 to see these new results.
> > W2. For certain tasks, G-ForgetNet performs worse than ForgetNet, or ForgetNet performs worse than the baseline. Some task-specific explanations regarding this should be discussed
>
> Generally ForgetNet struggles on algorithms where the $\texttt{hints}$ time series is very long, such as Jarvis' March, Articulation Points, or Heapsort, where the training time series lengths are 204, 163, and 91, respectively. This aligns with our expectations as it is incredibly difficult for the model to backpropagate signals through 100+ sequentially applied processor layers when the predictions early in the time series are inaccurate. We believe that ForgetNet underperforms on algorithms such as Naive String Matcher or Floyd-Warshall because the update step between consecutive $\texttt{hints}$ is non-trivial. E.g. in Floyd-Warshall, each step requires the model to simulate a double-nested for-loop, which is generally difficult to learn, and these difficulties are then compounded by the previously discussed training issues in ForgetNet.
>
> Conversely, ForgetNet performs best on algorithms with very simple update steps between hints such as in DFS.

---

> > ### Comment · Reviewer_gWrY · 2023-11-20
> >
> > Dear authors,
> >
> > Thanks for your response, but I'm afraid you have misunderstood my question. I'm asking if there is an inherent drawback of **neural algorithmic reasoning** in possessing the Markov property, rather than your proposed method. Could you provide any insights into this?

---

> ### Author Response · Authors · 2023-11-21
> **Reply to Reviewer gWrY**
>
> Dear reviewer gWrY,
>
> We apologize for any confusion. To the best of our knowledge, there are *not* any inherent drawbacks of neural algorithmic reasoning in possessing the Markov property. When the algorithmic reasoning task is Markov, neural algorithmic reasoners are able to learn single-step execution, *i.e.,* learning transitions between subsequent states or $\texttt{hints}$. As formalized in [1], intuitively, it is easier for neural networks to learn these simpler computations, and the removal of such Markov property would needlessly increase the complexity of the underlying task.
>
> Additionally, prior works such as [2] have shown that using the $\texttt{hints}$ time series significantly increases the performance of neural algorithmic reasoning models compared to directly learning to map from the algorithm input to the final output. They do not analyze this in the context of the Markov property, however, based on the analysis in our work, we can see that the removal of such hints removes the Markov property from the neural algorithmic reasoning task, which explains observed the performance degradation.
>
> [1] Keyulu Xu, Jingling Li, Mozhi Zhang, Simon S Du, Ken-ichi Kawarabayashi, and Stefanie Jegelka. What can neural networks reason about? In International Conference on Learning Representations, 2020.
>
> [2] Beatrice Bevilacqua, Kyriacos Nikiforou, Borja Ibarz, Ioana Bica, Michela Paganini, Charles Blundell, Jovana Mitrovic, and Petar Veličković. Neural algorithmic reasoning with causal regularisation. In International Conference on Machine Learning, pp. 2272–2288. PMLR, 2023.

---

> > ### Comment · Reviewer_gWrY · 2023-11-22
> >
> > Dear authors,
> >
> > Thanks for your clarification. Given your response and acknowledging my unfamiliarness with this area, I maintained my score and decreased my confidence.

---

### Official Review · Reviewer_fqct · 2023-10-31

**Soundness:** 3 good
**Presentation:** 3 good
**Contribution:** 3 good
**Rating:** 8
**Confidence:** 4

**Summary:**

This work studies a gap in the existing GNN literature -- while encoder-processor-decoder architectures typically have hidden states passed between calls to the processor, this work investigates the best way to handle those connections in the setting of algorithmic reasoning.

**Strengths:**

1. The work is original to my knowledge. In fact, I went through related work looking for experiments, claims, and ablation studies relevant to this paper. I expected to see some justification or discussion of the hidden states and the way they are used in other GNN architectures. I could not find a sufficient example and so I think question the architecture and showing that omitting hidden states all together or otherwise limiting the information flow through those connections is a valuable addition to the literature.
2. The quality of the writing is good. I think the paper is well motivated and clearly written.
3. The conclusions are significant. The fact is that on a 30-problem benchmark suite, the two architectural changes proposed in this work generally improve performance.

**Weaknesses:**

1. The discussion of the Markov nature of algorithms could use some nuance. If it is the case that these algorithms, when processing entire graphs, therefore having some history in the 'state' are Markov, then the historical information in the hidden features shouldn't help. In fact, this work shows that a gated layer is better than no connection for passing information with the hidden states. Thus, I think the story arc is a bit confusing. It actually took me several readings to really piece together the fact that the treatment of the hidden state is under discussed in the literature and the two options presented in this paper are better than the existing approaches as a result of the confusing Markov narrative. This is perhaps a small point, and I'm only one reader -- if the Markov framework helps convey the story to others this is my weakness, and not the paper's but I though it was worth mentioning here.

**Questions:**

1. In most algorithms ForgetNet beats the baseline (Figure 3). Is there any intuition or hypothesis around which particular algorithms the baseline is better? For example, for Naive String Matcher the baseline looks considerably better. Why might that be?
2. I really like the visualization with bars in Figure 1, I think adding G-ForgetNet to that graphic or otherwise including a plot with the G-ForgetNet results would help. Can the authors provide such a visualization?

---

> ### Author Response · Authors · 2023-11-20
> **Response to Reviewer fqct**
>
> Dear reviewer fpct, thank you for your valuable comments. Below are our responses to your concerns.
>
> > The discussion of the Markov nature of algorithms could use some nuance. If it is the case that these algorithms, when processing entire graphs, therefore having some history in the 'state' are Markov, then the historical information in the hidden features shouldn't help
>
> In our revised manuscript we have clarified our motivations for the G-ForgetNet model. The core motivation for the proposed G-ForgetNet is that, while inclusion of information from previous layers does not inherently align with the desired Markov nature, it can provide helpful support, especially during the early stage of training, where it can mitigate the effects of inaccurate intermediate predictions and facilitate higher quality backpropagation signals. *I.e.* the connection between layers in G-ForgetNet is beneficial not because the model is making use of historical information about the algorithm, but rather because of the important computational pathways it provides. We refer the reviewer to Section 4.1 and 4.2 for more elaboration. Additionally, we further introduce a loss penalty for G-ForgetNet and show that enforcing the gate to be closed further boosts performance, which aligns with our intuitions about the importance of obeying the Markov property. We hope that this new penalized G-ForgetNet model helps to clarify the story arc.
>
> > Q1.  In most algorithms ForgetNet beats the baseline (Figure 3). Is there any intuition or hypothesis around which particular algorithms the baseline is better? For example, for Naive String Matcher the baseline looks considerably better. Why might that be?
>
> Generally ForgetNet struggles on algorithms where the $\texttt{hints}$ time series is very long, such as Jarvis' March, Articulation Points, or Heapsort, where the training time series lengths are 204, 163, and 91, respectively. This aligns with our expectations as it is incredibly difficult for the model to backpropagate signals through 100+ sequentially applied processor layers when the predictions early in the time series are inaccurate. We believe that ForgetNet underperforms on algorithms such as Naive String Matcher or Floyd-Warshall because the update step between consecutive $\texttt{hints}$ is non-trivial. E.g. in Floyd-Warshall, each step requires the model to simulate a double-nested for-loop, which is generally difficult to learn, and these difficulties are then compounded by the previously discussed training issues in ForgetNet.
>
> > Q2.  I really like the visualization with bars in Figure 1, I think adding G-ForgetNet to that graphic or otherwise including a plot with the G-ForgetNet results would help. Can the authors provide such a visualization?
>
> We have added this visualization as Figure 6 in Appendix A.1.

---

> > ### Comment · Reviewer_fqct · 2023-11-21
> > **Reply from Reviewer fqct**
> >
> > Thank you for these additions. The paper is strong, and I advocate that it be accepted.

---

### Official Review · Reviewer_PbVk · 2023-11-01

**Soundness:** 3 good
**Presentation:** 2 fair
**Contribution:** 3 good
**Rating:** 6
**Confidence:** 3

**Summary:**

This paper proposes ForgetNet, a GNN that enforces Markov structure across algorithmic reasoning steps.
- Motivation: algorithmic tasks (e.g. sorting) can be modeled by transitioning through states which is Markov in nature, whereas existing methods use the history.
- Solution: ForgetNet enforces such Markov structure by updating the representation at each step usingfeatures at the current step only.
    - i.e. the hidden states at time $t$ is updated as $\{h_i^{(t)}\} = f_{\text{GNN}}(\{x_i^{(t)}\}, \{e_{ij}^{t}\}, g^{(t)})$, for node features $x$, edge features $e$, and global feature $g$.
- Further challenge: empirical results suggest that such Markov structure may cause training instability issues at early stage of training, likely because the intermediate steps are too unconstrained (note that supervision is provided only on the final state).
- Solution: G-ForgetNet, which selectively and adaptively keeps some history using a gating mechanism.

**Strengths:**

Another way to state the results is that
1. for better generalization, the authors proposed to restrict the function class by excluding non-Markov solutions, leading to the proposed ForgetNet.
2. the restricted function class raises optimization challenges. To address these challenges, the author incorporates a gating mechanism that selectively incorporate the history.

Both these changes are natural/simple yet effective: the proposed G-ForgetNet outperforms baselines in most tasks in the CLRS-30 benchmark.

**Weaknesses:**

- The discussion about the motivation could be improved.
- About the motivation that "algorithmic reasoning tasks are Markov": it should be clarified that this work is about algorithmic reasoning tasks that can be modeled by finite-state automata. In general, whether the process is Markov or not depends the definition of the state space.
- Even though the Markov observation motivates to remove history information, history information proves to be important for stabilizing training (hence the proposal of G-ForgetNet). This is similar to the effect of residual links, which can be discussed more in the paper.

**Questions:**

- Fig 4 (b): what are execution steps, and how are the labels acquired? Since there are labels to compute the per-step loss, can we also compare with per-step teacher forcing?
- Fig 5: for better comparison, could you also provide the norm of the residual branch in the baseline?
- I wonder if one can anneal the gating, i.e. gradually reducing the amount of the history (e.g. by controlling the gating value).
    - One motivation is that in Table 1, when G-ForgetNet is not better than the baseline, its performance is very close to the baseline. This suggests that G-ForgetNet falls back to the baseline in these cases, hence ensuring the history embeddings are actually not being used may help with performance.
    - Another reason is that according Fig 5, the gate value is ~0.38 even at the end of training. This means the history embeddings are still helpful, which weakens the claim that the task should be Markov, unless the history embeddings are effectively removed.

---

> ### Author Response · Authors · 2023-11-20
> **Response to Reviewer PbVk 1/2**
>
> Dear reviewer PbVk,
>
> We thank you for your insightful comments. We have addressed your comments in the revised manuscript.
>
> >  The discussion about the motivation could be improved.
>
> Prior works such as [1] have shown the importance of aligning the computational graph of neural networks with the structure of the underlying task. This motivates us to restructure the computational graph of neural algorithmic reasoners to be better aligned with the Markov nature of the algorithms in the CLRS-30 benchmark. In our revised manuscript we have clarified the motivation for our G-ForgetNet model. We refer the reviewer to Sections 4.1 and 4.2 to see our refined motivation for G-ForgetNet, including the mentioned connections with residual links.
>
> >  It should be clarified that this work is about algorithmic reasoning tasks that can be modeled by finite-state automata. In general, whether the process is Markov or not depends the definition of the state space.
>
> Firstly, thank you for raising an interesting perspective on finite-state automata, which may broaden this direction in the future. Indeed, the algorithms in the CLRS-30 *can* be modeled by finite automata under a slightly irregular definition of the state space because the graphs used for training and testing are of fixed size. We leave further theoretical analysis of algorithmic reasoning in the context of finite automata or other models of computation to future work.
>
> >  Even though the Markov observation motivates to remove history information, history information proves to be important for stabilizing training (hence the proposal of G-ForgetNet). This is similar to the effect of residual links, which can be discussed more in the paper.
>
> We appreciate the insightful comments about the similarities with residual links. We have added some discussion on these similarities in Section 4.1.
>
> > Fig 4 (b): what are execution steps, and how are the labels acquired? Since there are labels to compute the per-step loss, can we also compare with per-step teacher forcing?
>
> This part of Figure 4 was moved to Appendix B.2 in our updated manuscript in order to make room for the rest of our updates. However, the execution steps in this figure are the time series of $\texttt{hints}$ for each intermediate step in the algorithm process. Per your suggestion, we evaluated ForgetNet with 100% teacher forcing. Allthough the losses for ForgetNet with teacher forcing are incredibly low, the model fails to generalize to the test data for most algorithms. The figure with the training losses for ForgetNet trained with teacher forcing on the Floyd-Warshall algorithm can be sound in the Supplementary Material, and the performance for ForgetNet with teacher forcing on each algorithm can be found below. Allthough the overall average performance of ForgetNet with teacher forcing is significantly below the baseline, it outperforms our best model, G-ForgetNet, on several algorithms, including naïve string matcher, where it achieves a remarkable 100% test accuracy. We believe teacher forcing shows significant potential for future works, however further study is outside the scope of our paper.
>
> > Fig 5: for better comparison, could you also provide the norm of the residual branch in the baseline?
>
> Yes, per your suggestion, we have included the norm of the residual branch in the baseline in Figure 5 as well as in Figure 7 in Appendix A.2. In order to provide a fair comparison between G-ForgetNet and the baseline, we now report the norm of the gated hidden states instead of the norm of the gate itself.
>
> [1] Keyulu Xu, Jingling Li, Mozhi Zhang, Simon S Du, Ken-ichi Kawarabayashi, and Stefanie Jegelka. What can neural networks reason about? In  International Conference on Learning Representations,  2020.

---

> > ### Author Response · Authors · 2023-11-20
> > **Response to Reviewer PbVk 2/2**
> >
> > > I wonder if one can anneal the gating, i.e. gradually reducing the amount of the history (e.g. by controlling the gating value).
> > > -   One motivation is that in Table 1, when G-ForgetNet is not better than the baseline, its performance is very close to the baseline. This suggests that G-ForgetNet falls back to the baseline in these cases, hence ensuring the history embeddings are actually not being used may help with performance.
> > >- Another reason is that according Fig 5, the gate value is ~0.38 even at the end of training. This means the history embeddings are still helpful, which weakens the claim that the task should be Markov, unless the history embeddings are effectively removed.
> >
> > Firstly, I believe was have addressed the concerns here by adding the gate penalty to G-ForgetNet and showing that is does in fact go to ~0 at the end of training, thereby enforcing the Markov property. Additionally, there are no longer any cases where G-ForgetNet is not better than the baseline.
> > Secondly, we had previously tried a manually annealed schedule and found that it underperformed ForgetNet, performing about the same as the baseline, so we did not include these results in the paper. The performance on each algorithm for the annealed model is also reported in the table below.
> >
> > ### Performance of requested Teacher Forcing and Annealed models
> > Results outperforming G-ForgetNet are highlighted in bold
> > |Algorithm| ForgetNet - Teacher Forcing | Manual Annealing |
> > |--|--|--|
> > | Activity Selector | 91.89±2.20% | 95.99±1.02% |
> > | Articulation Points | 10.75±3.17 | 94.37±0.84% |
> > | Bellman-Ford | 95.96±0.30% | 98.71±0.08% |
> > |BFS | **99.98±0.02%** | 99.95±0.02% |
> > |Binary Search | 65.42±1.60% | 71.40±2.22% |
> > | Bridges | 22.34±3.93% | 96.25±1.41% |
> > | Bubble Sort | 9.28±0.86% | 77.53±3.20% |
> > | DAG Shortest Paths | 60.64±1.08% | 98.45±0.17% |
> > | DFS | 19.00±0.80% | 62.23±7.53% |
> > | Dijkstra | 64.74±4.20% | 98.23±0.06% |
> > | Find Max. Subarray | **80.97±1.03%** | 42.89±0.98% |
> > | Floyd-Warshal | 5.91±0.13% | 33.77±1.15% |
> > | Graham Scan | 66.82±2.23% | 92.17±1.41% |
> > | Heapsort | 6.01±0.61% | 37.26±3.89% |
> > | Insertion Sort | 64.13±1.57% | 87.52±0.93% |
> > | Jarvis' March | **95.97±0.38%** | 82.45±1.45% |
> > | Knuth-Morris-Pratt | 14.43±1.05% | **23.79±2.96%** |
> > | LCS Length | 71.87±8.48% | 84.49±0.60% |
> > | Matrix Chain Order | 78.72±0.59% | 92.94±1.08% |
> > | Minimum | 99.01±0.14% | 97.73±0.40% |
> > | MST-Kruskal | 15.60 ±0.93% | 85.73±0.72% |
> > | MST-Prim | 69.20±2.34% | 92.75±0.53% |
> > | Naïve String Matcher | **100.00±0.00%** | 27.32±1.39% |
> > | Optimal BST | 36.60±5.64% | 78.52±0.73% |
> > | Quickselect | 1.99±0.12% | 3.54±0.90% |
> > | Quicksort | 8.47±0.80% | 49.56±5.09% |
> > | Segments Intersect | 99.01±0.09% | 98.47±0.06% |
> > | SCC | 8.94±1.90% | 41.90±2.54% |
> > | Task Scheduling | 79.98±3.72% | 81.05±0.44% |
> > | Topological Sort | 3.65±0.29% | 91.48±3.68% |
> > | Overall Average | 50.47±1.97% | 73.95±1.58% |

---

> > > ### Comment · Reviewer_PbVk · 2023-11-22
> > >
> > > Thank you very much for the clarification and additional results!
> > > I maintain my score and recommend acceptance.

---

### Official Review · Reviewer_VNPs · 2023-11-10

**Soundness:** 3 good
**Presentation:** 3 good
**Contribution:** 2 fair
**Rating:** 6
**Confidence:** 4

**Summary:**

The paper focuses on the Markov property of neural algorithmic reasoning. More specifically, neural networks that learn to imitate algorithmic execution have thus far used as additional inputs data from the algorithmic traces at previous execution steps, not just the current step. The paper emphasises that, as the studied algorithms are Markovian, a better alignment between the neural network and the task should mean not using data from previous execution steps, which are named historical embeddings in the paper. This work uses an established architecture as the baseline, and with the removal of the historical embeddings proposes ForgetNet. Further, G-ForgetNet is obtained by using a gating mechanism to learn how much to use the historical embeddings within the baseline. The proposed changes outperform the studied baselines, and a simple ablation study on the gating mechanism shows the importance of using historical embeddings early on in training, when accumulating errors can be more harmful for final performance.

**Strengths:**

The paper is well motivated and clear. The proposed changes are supported by the empirical evaluation on the CLRS-30, outperforming the baseline in most algorithms.

**Weaknesses:**

The initial hypothesis — that better alignment with the markovian property should result in better performance — is strong. However, there are trainability issues (e.g. accumulating errors early in training) that can be addressed in different ways. The paper proposes one solution, gating, but some questions remain — why is gating better than the NN more generally learning how to combine historical and current embeddings, as in the baseline? Moreover, what are other solutions, possibly not involving learning?

Lastly, the TripletGMPNN includes a gating layer in the single-task setup. A discussion on how this differs from the gating in G-ForgetNet should be included.

**Questions:**

Please see above.

---

> ### Author Response · Authors · 2023-11-20
> **Response to Reviewer VNPs**
>
> Dear reviewer VNPs, thank you for your valuable comments. Below are our responses to your concerns.
>
> >  Why is gating better than the NN more generally learning how to combine historical and current embeddings, as in the baseline?
>
> In response to your comments and those of other reviewers, we have added a regularization term to the loss function which penalizes the magnitude of the gate norm, ensuring that the gate is aligned with the Markov property as desired. However, to answer your original question, the G-ForgetNet model *without* this penalty performs better than the baseline because, while it still may fit some historical noise, it is less reliant on historical embeddings than the baseline model, and therefore closer aligned with the Markov property. We further analyze your question in Appendix A.3 of our revised manuscript.
>
> >  Lastly, the TripletGMPNN includes a gating layer in the single-task setup. A discussion on how this differs from the gating in G-ForgetNet should be included.
>
> The gate in Triplet-GMPNN is placed ahead of the decoder and is meant to bias the model towards only updating a subset of the hidden states at each step, while the remaining hidden states are left unchanged. Our gate is placed ahead of the processor and is meant to bias the model towards not including *any* hidden states at all, which we further enforce using a loss penalty. We have added further clarification about the difference between our gates in Section 5 of the revised manuscript.
>
> > Moreover, what are other solutions, possibly not involving learning?
>
> We previously tried to use a manual annealing schedule for the gate in which the gate is initially fully open, and is linearly decayed until the end of training, where it would be entirely closed, thus explicitly enforcing the Markov property. However, we found that this did not significantly improve the performance over the baseline. The results of this experiment for each algorithm are given in the table below.
>
> Results outperforming G-ForgetNet are highlighted in bold
>
> |Algorithm | Manual Annealing |
> |--|--|
> | Activity Selector | 95.99±1.02% |
> | Articulation Points | 94.37±0.84% |
> | Bellman-Ford | 98.71±0.08% |
> |BFS |  99.95±0.02% |
> |Binary Search | 71.40±2.22% |
> | Bridges | 96.25±1.41% |
> | Bubble Sort | 77.53±3.20% |
> | DAG Shortest Paths | 98.45±0.17% |
> | DFS |  62.23±7.53% |
> | Dijkstra | 98.23±0.06% |
> | Find Max. Subarray | 42.89±0.98% |
> | Floyd-Warshal | 33.77±1.15% |
> | Graham Scan | 92.17±1.41% |
> | Heapsort | 37.26±3.89% |
> | Insertion Sort | 87.52±0.93% |
> | Jarvis' March | 82.45±1.45% |
> | Knuth-Morris-Pratt | **23.79±2.96%** |
> | LCS Length |  84.49±0.60% |
> | Matrix Chain Order | 92.94±1.08% |
> | Minimum | 97.73±0.40% |
> | MST-Kruskal | 85.73±0.72% |
> | MST-Prim | 92.75±0.53% |
> | Naïve String Matcher | 27.32±1.39% |
> | Optimal BST | 78.52±0.73% |
> | Quickselect | 3.54±0.90% |
> | Quicksort | 49.56±5.09% |
> | Segments Intersect | 98.47±0.06% |
> | SCC | 41.90±2.54% |
> | Task Scheduling | 81.05±0.44% |
> | Topological Sort | 91.48±3.68% |
> | Overall Average | 73.95±1.58% |

---

> > ### Comment · Reviewer_VNPs · 2023-11-23
> >
> > I would like to thank the authors for their response. I will maintain my score, recommending acceptance of the paper.

---

### Author Response · Authors · 2023-11-20
**Revisions Summary**

Dear reviewers, thank you for your valuable comments and insights.

Firstly, in response to comments and concerns raised by multiple reviewers about the ability of G-ForgetNet to be consistent with the Markov property, we have added a regularization term to the loss function which penalizes the magnitude of the gate norm. This change is supported by strong empirical results and an in-depth analysis of the gate during training. Specifically, we show that the gate magnitude converges to 0 over the course of training, and as such, the model is well aligned with the Markov property during testing. With this change, G-ForgetNet outperforms the baseline on all 30/30 algorithms and surpasses the 99% accuracy threshold on 9/30 algorithms, compared to just 1/30 in the baselines. **All of the changes in the revised manuscript are highlighted in red to make it clear which parts were updated.**

Secondly, we have added an experiment in Appendix B.1 which evaluates the benefits of the Markov property in the multi-task setting of [1]. As with the single-task ForgetNet experiments in Section 3.2, we observe that explicitly enforcing the Markov property of algorithmic reasoning is beneficial for model generalization, as the multi-task ForgetNet outperforms the multi-task baseline from [1] on 20/30 of the CLRS-30 algorithms.

Finally, we would like to highlight a mistake in the original submission that has since been fixed in the revised manuscript. The original submission reported the standard deviation between runs in Table 1 and for G-ForgetNet in Table 2, however the rest of the models in Table 2 and the previous literature report the standard error of the mean. We have corrected all the reported values to use the standard error of the mean to be consistent with previous works.

[1]. Borja Ibarz, Vitaly Kurin, George Papamakarios, Kyriacos Nikiforou, Mehdi Bennani, Róbert Csordás, Andrew Joseph Dudzik, Matko Bošnjak, Alex Vitvitskyi, Yulia Rubanova, et al. A generalist neural algorithmic learner. In Learning on Graphs Conference, pp. 2–1. PMLR, 2022.

---

### Meta-Review · Area_Chair_Aev8 · 2023-12-04

**Metareview:**

This paper introduces a Markov condition in neural algorithmic reasoning, providing an inductive bias that is shared by many algorithmic reasoning tasks. Overall, the method is innovative, with clear improvements. The choice of the gating mechanism for selecting the historical embedding is only one of many choices. This was called out as a negative of the paper. At the same time, I believe that this could spark future contributions. I recommend the authors clearly mention these directions for future improvements and address the other comments by reviewers (e.g. connection with finite state automata).

**Justification For Why Not Higher Score:**

The reviewers still had issues with the scalability of the proposed methodology. ForgetNet is hard to train, which is why relaxed versions end up performing better in practice.

**Justification For Why Not Lower Score:**

The idea is elegant and novel, likely to inspire new results if highlighted at the conference.

---

### Decision · Program_Chairs · 2024-01-16

Accept (spotlight)